# VLDLR mediates Semliki Forest virus neuroinvasion through the blood-cerebrospinal fluid barrier

Miika Martikainen [1] ✉, Roberta Lugano[1,2], Ilkka Pietilä[1,2], Sofie Brosch[1], Camille Cabrolier[1], Aishwarya Sivaramakrishnan[1], Mohanraj Ramachandran [1], Di Yu [1], Anna Dimberg [1] & Magnus Essand [1]

Semliki Forest virus (SFV) is a neuropathogenic alphavirus which is of interest both as a model neurotropic alphavirus and as an oncolytic virus with proven potency in preclinical cancer models. In laboratory mice, peripherally administered SFV infiltrates the central nervous system (CNS) and causes encephalitis of varying severity. The route of SFV CNS entrance is poorly understood but has been considered to occur through the blood-brain barrier. Here we show that neuroinvasion of intravenously administered SFV is strictly dependent on very-low-density-lipoprotein receptor (VLDLR) which acts as an entry receptor for SFV. Moreover, SFV primarily enters the CNS through the blood-cerebrospinal fluid (B-CSF) barrier via infecting choroid plexus epithelial cells which show distinctly high expression of VLDLR. This is the first indication of neurotropic alphavirus utilizing choroid plexus for CNS entry, and VLDLR playing a specific and crucial role for mediating SFV entry through this pathway.

Semliki Forest virus (SFV) is an alphavirus, i.e., a group IV, positive-sense single-stranded RNA [+ssRNA] virus in the Togaviridae family. SFV has been studied extensively as a model neurotropic alphavirus as well as an immunotherapeutic agent in preclinical tumor models including glioblastoma[1–3]. In vitro, SFV infects and replicates in a notably wide range of cell types, originating from different hosts species. In mice SFV shows very clear tropism towards the central nervous system (CNS), in particular towards oligodendrocytes and neurons, and causes encephalitis[4].

The model for SFV-induced neuropathology in laboratory mice is well established. Previously published work show that intravenous or intraperitoneal injections of SFV in laboratory mice lead to viral invasion of the central nervous system (CNS) and neuropathology of varying severity, depending on the SFV strain and age of the mice[5–7]. Neurovirulent SFV strains (such as SFV4) can readily infect neurons in adult mice, while avirulent strains (such as A774) can only infect neurons in young mice. The subsequent neuronal damage caused by the virus and the consequent inflammation leads to neurological symptoms such as hyperexcitability, convulsions, ataxia and paralysis. The difference in neuronal replication between A774 and SFV4 is linked to difference in viral non-structural protein sequence, with non-structural protein 3 (nsP3) being the major neurovirulence factor in SFV4[7]. SFV neuroinvasion is generally considered to occur through the blood-brain barrier (BBB) endothelial cells[8], but other infiltration routes cannot be ruled out. Of note, while the envelope glycoproteins of both A774 and SFV4 can mediate CNS invasion, the spike of otherwise avirulent A774 shows better penetration in an in vitro BBB model[9] and increased pathogenicity in vivo[7].

Recently, the Very Low-Density Lipoprotein Receptor (VLDLR) and Apolipoprotein E receptor 2 (ApoER2, encoded by the Lrp8 gene) were identified as entry receptors for Semliki Forest virus (SFV), Sindbis virus (SINV) and Easter equine encephalitis virus (EEEV)[10]. The structure of SFV in complex with VLDLR has also been solved and indicates that SFV E1-DIII sites on the virion spike enable binding to VLDLRs from divergent host species[11]. VLDLR is highly conserved and widely expressed which explains the wide in vitro tropism of SFV. Both

[1]Department of Immunology, Genetics and Pathology, Science for Life Laboratory, Uppsala University, Uppsala, Sweden. [2]These authors contributed equally: Roberta Lugano, Ilkka Pietilä. ✉e-mail: miika.martikainen@igp.uu.se

VLDLR and ApoER2 are expressed in mouse CNS, which correlates with SFV being infectious in CNS tissue. However, the role of VLDLR and/or ApoER2 in mediating SFV neuroinvasion from peripheral tissues requires further investigation.

Here, we show that the two SFV prototype strains A774 (avirulent) and SFV4 (neurovirulent) exhibit difference in their VLDLR dependency in vitro, and that this is due to a difference in their structural protein sequence. Our in vivo results demonstrate that SFV entry into the CNS occurs through the blood-cerebrospinal fluid (B-CSF) barrier at the choroid plexus (CP). The choroid plexus epithelial cells (CPEpiCs) show high basal expression of VLDLR. The lack of VLDLR expression in *Vldlr*[-/-] mice renders SFV unable to invade the CNS from the circulation. Our results therefore pinpoint an important role of VLDLR in mediating neuroinvasion of SFV.

## Results

### CRISPR library screen identifies VLDLR as SFV receptor

To identify the SFV entry receptor, we used the human genome-wide Brunello CRISPR/Cas9 knockout library[12] in a screen on human osteosarcoma cells (HOS, ATCC, CRL-1543) infected with fully replicative SFV4. While these cells are not a natural target for neurotropic SFV, they offer a highly SFV-permissive and easy to culture model needed for the screen. We performed the screening both with and without the janus kinase (JAK) inhibitor ruxolitinib as we hypothesized that ruxolitinib would limit the host cells capability to resist SFV replication and therefore sensitize the screen more towards factors associated with viral entry. Schematic of the screen design is shown in Fig. 1a.

As expected, the screening with ruxolitinib resulted in stronger cytopathic effect in target cells (Fig. S1a). This also correlated with differently skewed screening result. Screening without ruxolitinib resulted in only 4 enriched hits (false discovery rate (FDR) < 0.05); *ATP6V0B*, *ATP6AP1*, *ATP6V0C* and *ATP6V1B2* (Fig. S1b), all of which can be linked to endosomal escape of the virus, but unlikely have direct entry receptor function. Results from the screening without ruxolitinib also reveal five significantly depleted (implying anti-viral effect) gene hits. With ruxolitinib the number of enriched hits increased to 24. In line with the report by Clark et al. [10], *VLDLR* was one of the potential entry receptor hits (Figs. 1b and S1b). A list of all significant hits with a false discovery rate (FDR) < 0.05 is presented in Fig. S1b. A full list of the results is provided as tables in Supplementary data 1 and 2.

### SFV4 shows strong VLDLR-dependency and A774 is not strictly VLDLR-dependent

We next evaluated if VLDLR is a general receptor for SFV strains with different structural spike glycoproteins (E1-E2 trimer, Fig. 1c). For this, we selected SFV4 and the avirulent A774 strain (Fig. 1d), which differ in the structural spike glycoproteins E1 and E2 and (Fig. S2a). Sequence alignment with other common SFV strains SFV6, L10, and A7, shows that SFV4 and A774 also capture the general sequence variation observed across these strains (Fig. S2a). In addition, we used the previously described chimeric A774-V4nstr strain[7] (herein labeled as SFV4(A774st)), which has the replication-affecting non-structural genes of SFV4 and the structural glycoprotein genes from A774 (Fig. 1d). This chimeric strain therefore allows identification of differences in infectivity that are specifically related to structural proteins, and therefore likely caused by differences in entry.

Consistent with SFV4 entry being VLDLR mediated, blocking VLDLR ligand-binding LDLR class A (LA) repeats with a monoclonal antibody (mAb)[13] protects HOS cells from SFV4-mediated lysis as compared to an isotype control antibody (Fig. 1e). VLDLR blocking gave no significant protection against A774 or SFV4(A774st) (Fig. 1e). This indicates that, in contrast to the SFV4 strain, entry of SFV strains carrying A774 structural proteins is not strictly VLDLR-dependent. Comparison of different VLDLR blocking mAbs clones show that 1H5

(binding LA repeats 2, 5 and 6[13]) and 1H10 (binding repeats LA 3, 4, 5, and 6[13]) inhibited SFV4 killing (Fig. S2b, c). In contrast, mAb 5F3 (binding LA repeats 7, 8) gave no significant protection (Fig. S2b, c). Our results are in line with recent results by Cao et al., showing that VLDLR LA repeats 1, 2, 3, and 5 synergistically participate in SFV binding[11]. Infecting HOS cells with fluorescent reporter SFV4 (SFV4-dsRed) resulted in markedly lower replication (as detected by fluorescence intensity with plate reader) in cells pre-incubated with mAb 1H10 or 1H5 (Fig. S2d). In line with A774 entry not being solely dependent on VLDLR, none of the mAbs used gave significant protection against A774 infection (Fig S2c) and showed no reduction in fluorescent reporter A774-mCherry replication (Fig. S2d).

The VLDLR-negative K562 cell line supported infection of both A774 and SFV4(A774st) resulting in >100-fold higher virus titers in the culture medium as compared to SFV4 at the d2 time point post infection (Fig. 1f). This further reinforces the lower VLDLR-dependency of A774. In K562 cells engineered to express VLDLR (K562-VLDLR, Fig. 1g), titers of all viruses increased as compared to wild-type K562 cells (Fig. 1f). Based on these results we conclude that SFV constructs carrying A774 structural proteins can use VLDLR as a receptor but can readily also utilize VLDLR-independent entry. On the other hand, entry of viruses with SFV4 structural proteins is dependent on VLDLR in all tested cell lines.

### VLDLR is distinctly highly expressed in choroid plexus epithelial cells

To elucidate the role of VLDLR for SFV in vivo, we first analyzed VLDLR expression in the mouse brain with specific focus on possible CNS entry points. Immunofluorescence (IF) staining shows that, in adult C57BL/6J mouse brain, VLDLR is highly expressed in the choroid plexus (CP) epithelial cells (CPEpiC) (Fig. 2a, b) which can be identified by the anatomical structure and expression specific markers such as AQP1 (Fig. 2c). This finding is also supported by available single cell RNA sequencing (scRNA-seq) data[14] (Fig. 2d). Furthermore, available scRNA-seq[15,16] data also indicates low or lack of *Vldlr* expression in brain endothelial cells (Figs. 2d and S3a). In line with this, VLDLR staining was not observed in vascular structures in the brain of WT mice (Fig. S3b). Notably, Both A774 and SFV4(A774st) were capable of infecting primary mouse brain endothelial cells in vitro whereas SFV4 was not (Fig. S3c). This supports that the A774 structural proteins can mediate VLDLR-independent entry not only in HOS and K562 cell lines, but also in mouse brain endothelial cells.

ApoER2 (encoded by the *Lrp8* gene) is another low-density lipoprotein receptor which can function as an entry receptor for SFV[10] and compared to *Vldlr*, a lower percentage of CPEpiCs show *Lrp8* expression (Fig. 2d, g). Further, CPEpiCs show low level ApoER2 staining with a clear polarization to the surface facing the CSF fluid (Fig. 2e). Consistent with this, apical localization of ApoER2 has been reported in chicken CPEpiCs[17]. ApoER2 and VLDLR protein can also be found with staining in cells lining the ventricle walls (Fig. 2f), in line with expression of *Vldlr* and *Lrp8* in ependymocytes (Fig. 2d). In contrast to *Vldlr*, *Lrp8* expression can be seen in brain endothelial cells (Figs. 2d and S3b) with the previously reported abluminal localization in capillary endothelial cells[18]. There is however no clear indication of ApoER2 protein or *Lrp8* gene expression in choroid plexus endothelial cells (Fig. 2e, g). Taken together, these data suggest that from the two previously reported SFV receptors VLDLR and ApoER2, only VLDLR is abundantly present on the basolateral membrane of CPEpiCs.

### Early SFV neuroinvasion occurs through choroid plexus epithelial cells

As reported previously[7], SFV4 and SFV4(A774st) are both pathogenic in mice, with neurological symptoms (humane end point) appearing from 4 to 6 days after $1 \times 10^6$ PFU IV injection (Fig. 3a). The onset of symptoms is always linked to a clear presence of viral proteins in the

brain parenchyma (Fig. 3b), and notable increase of immune cells (Fig. S4a–d), including CD4[+] and CD8[+] T cells (CD45[+]/CD11b[-]) and CD45[+]/CD11b[+] myeloid cells (Fig. S5). Taken together, this indicates severe SFV-induced immune-associated neuropathology.

To visualize the early kinetics of SFV in the brain following systemic infection, we collected different brain regions (Fig. 3c) and measured the presence of fully replicative virus (plaque titration) at day 2, 3, and 4 after SFV4(A774st) IV injection. Blood and cerebrospinal fluid (CSF) samples were also analyzed by plaque titration at the same time points. Infectious virus was detected in olfactory bulb, cortex, midbrain and cerebellum samples (Fig. 3c). Virus was also found in blood samples until day 3 (Fig. 3c). The transient viremia followed by CNS invasion matches well with previously reported studies[8]. Virus was detected also

in 2/5 CSF samples collected from WT mice on day 3 (Fig. 3c). This suggests that the virus has access to the CSF relatively early after intravenous virus injection. To increase the probability of detecting the virus from the very small volume CSF samples, we injected additional mice with higher dose of the virus (1 × 10[7] PFU) and analyzed the CSF. While this did not lead to higher proportion of samples being positive, we could detect higher titers in the positive samples (Fig. 3c). Further supporting the virus reaching CSF from the circulation, we can detect viral proteins in CPEpiCs (Fig. 3d) and cells lining the ventricle walls (Fig. 3e) as early as day 2 after IV virus injection. The CSF has been shown to naturally flow from the ventricular system ventrally to the basal cisterns and under the olfactory bulbs[19] (Fig. 3f). Matching this flow pattern, SFV protein was detected in the high VLDLR-expressing regions in

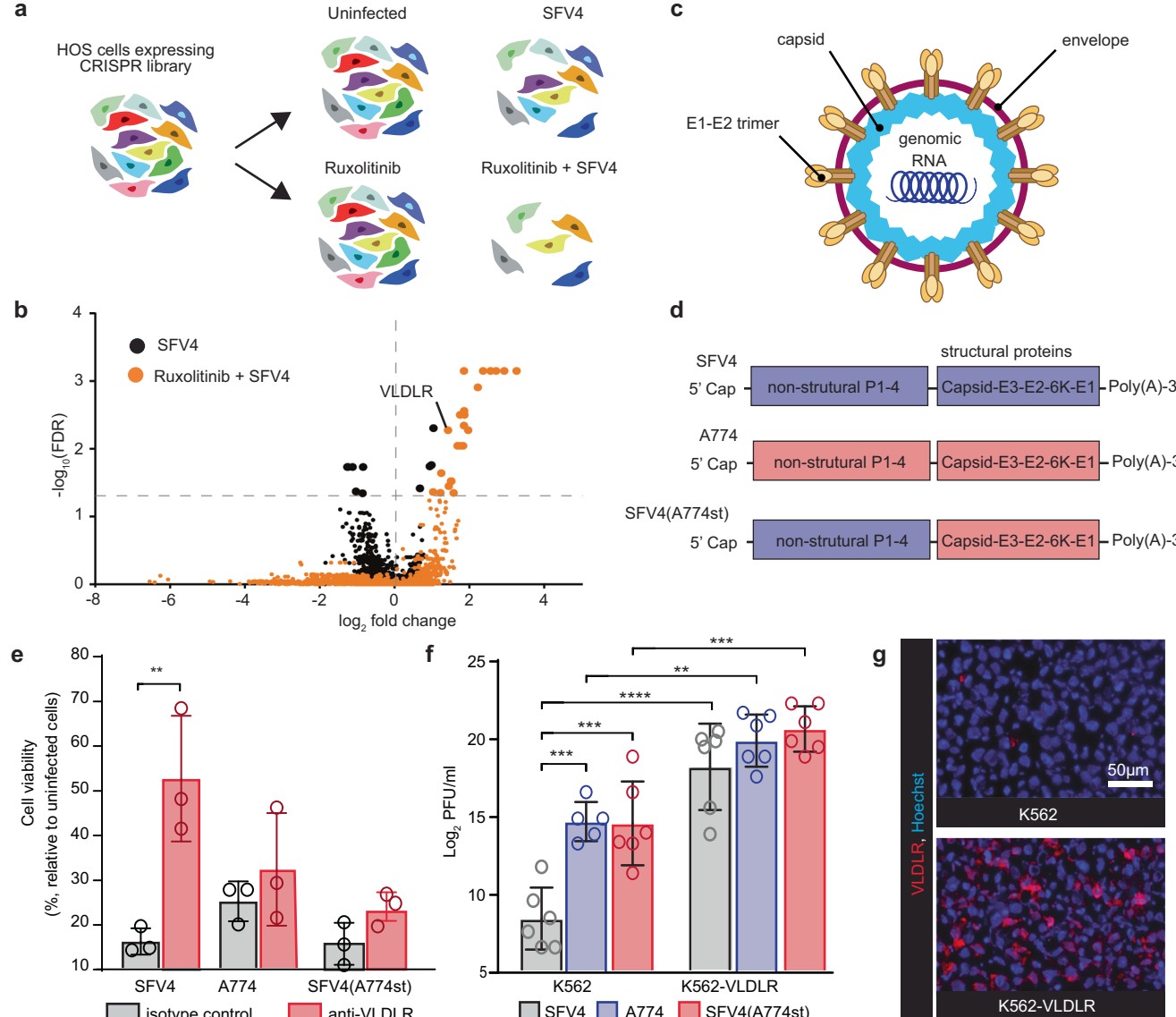

**Fig. 1 | A genome-wide CRISPR/Cas9 screen identifies VLDLR as entry receptor for SFV4. a** Schematic presentation of the CRISPR/Cas9 screen experiment. **b** Results of the MAGeCK analysis. Statistically significant hits indicated with bigger dots. **c** Schematic illustration of the SFV virion. **d** Schematic illustration of the used SFV constructs. **e** Incubating HOS cells with VLDLR-blocking antibody reduces SFV4 infection, but not A774 or SFV4(A774st) (structural A774 proteins) infection. Cell viability of infected (MOI = 0.1) HOS cells pre-incubated with the LDLR class A-specific mAb 1H5, or isotype control, in an attempt to block SFV entry. Cell viability measured with MTT-assay 48 h after infection. Data plotted as mean

(n = 3) ± SD. **f** Titration (PFU/ml) from cell culture medium collected 48 h after infection of K562 cells and K562 cells engineered to express VLDLR (K562-VLDLR) with SFV4, A774 or SFV4(A774st) (MOI = 0.01). Data presented in Log$_2$ scale for statistical analysis. Data plotted as mean (n = 6, for K562 infected with A774 n = 5) ± SD. **g** Immunofluorescence staining showing VLDLR expression in K562-VLDLR. Each datapoint in (**e**, **f**) represents the result from an independent biological replicate. Statistical analysis is done using One-way ANOVA with Tukey's multiple comparisons test. Source data and exact *P* values are provided as a Source Data file. PFU plaque forming units, MOI multiplicity of infection.

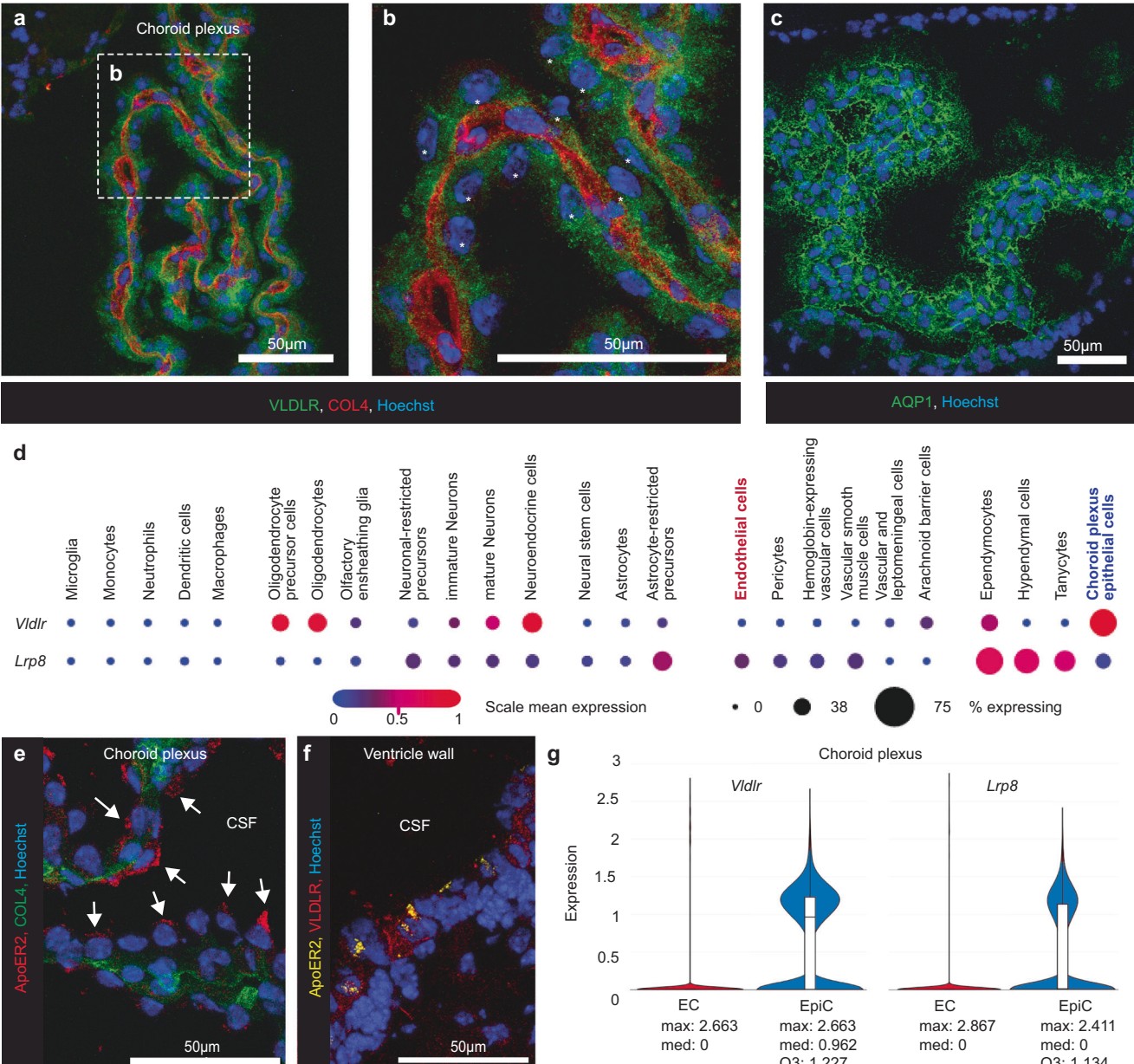

**Fig. 2 | VLDLR is expressed in choroid plexus epithelial cells.** Staining for VLDLR and basement membrane (Type IV collagen, COL4) in the choroid plexus in C57BL/6 J mice (**a**, **b**) show clear expression of VLDLR in epithelial cells. **b** Magnified insert from (**a**). Epithelial cell nuclei indicated with white asterisks. **c** staining for choroid plexus marker aquaporin-1 (AQP1). **d** single cell RNAseq data (reproduced from[14]) showing expression of *Vldlr* and *Lrp8* (gene encoding ApoER2) in different cell types in 2–3 month old C57BL/6 mouse brain. Choroid plexus epithelial cells (red) show

*Vldlr*[high]/*Lrp8*[low] expression profile. **e** immunostaining in choroid plexus shows apical ApoER2 polarization in epithelial cells (indicated with arrows). **f** Immunostaining for VLDLR and ApoER2 in ventricle wall. CSF: cerebrospinal fluid. **g** single cell RNAseq violin plot (reproduced from ref. 34) of *Vldlr* and *Lrp8* in choroid plexus endothelial cells (EC) and epithelial cells (EpiC) in adult mice. Maximum (max), median (med) and third quartile (Q3) values are provided below the graphs.

the ventral side of the brain (near the olfactory bulb) at day 4 after IV SFV4(A774st) injection (Fig. 3f). This occurs before the onset of symptoms, which suggests that SFV first gains access to the ventricles and subsequently spread to other regions in the brain. Negative controls for VLDLR and SFV staining are shown in Fig. S6.

SFV4(A774st) infected C57BL6/J mice show signs of BBB damage, as indicated by leakage of IV injected cadaverine, as a small-molecule (950 Da) fluorescent probe, into the brain parenchyma at day 4 after virus injection (Fig. S7a–e). Viral proteins can also be detected in CD31[+] endothelial cells at the time point prior to onset of leakage (Fig. S7c). The early presence of virus in CSF and CPEpiCs, low penetration of BBB and relatively late appearance of leaky BBB points towards SFV gaining initial early entry into CNS via B-CSF barrier rather than the BBB.

## VLDLR is crucial for SFV neuroinvasion from the circulation

To elucidate the role of VLDLR for SFV in vivo, we studied SFV pathology in homozygous Vldlr knock-out mice (B6;129S7-*Vldlr*[tm1Her]/J, Vldlr KO) in comparison to B6129SF2/J control mice. While some background staining for VLDLR can be detected in the brains of *Vldlr* KO mice (especially in the cerebellum, Fig. S6) they lack expression of VLDLR in the choroid plexus cells (Fig. 4a, b). *Vldlr* KO mice showed notable resistance to IV injected SFV4 and SFV4(A774st) with a majority (9/10) of mice not succumbing to neurotoxicity (Fig. 4c). We decided to employ SFV4(A774st) for our further studies due to its ability to utilize both VLDLR-dependent and independent entry therefore allowing relevant conclusion to be made about the importance of disrupting VLDLR-dependent entry in vivo.

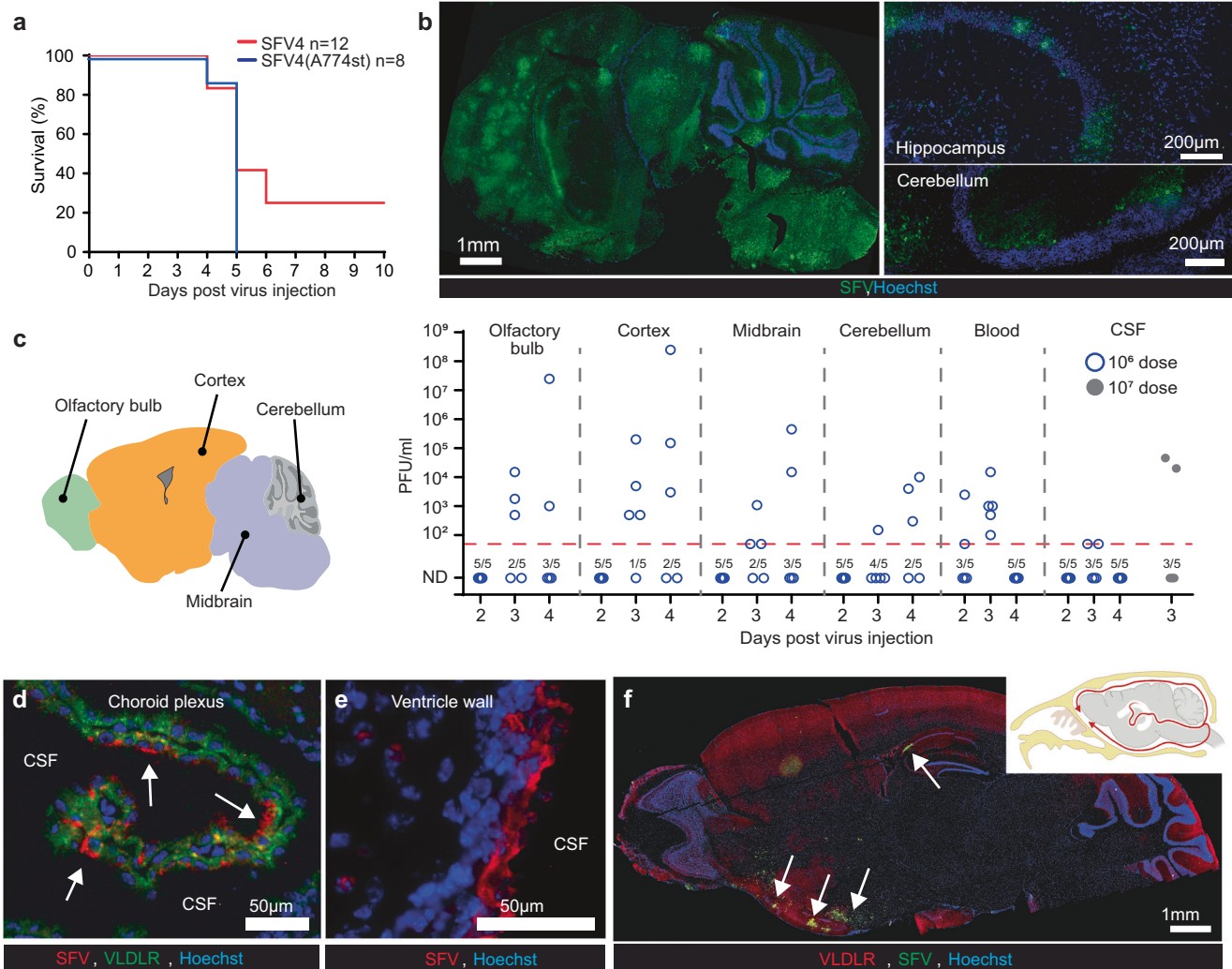

**Fig. 3 | Early SFV neuroinvasion in C57BL/6J mice occurs through choroid plexus epithelial cells. a** Kaplan–Meier analysis of mice infected intravenously (IV) with $1 \times 10^6$ PFU SFV4 or SFV4(A774st). Statistical analysis done with log-rank test. Data from two independent experiments. **b** Representative immunostaining for SFV proteins in mouse brain at end point (day 5) after SFV4(A774st) injection. Magnified regions are shown from hippocampus and cerebellum. **c** Analysis of virus in different brain regions (as indicated), blood and CSF with plaque titration (PFU/ml). Samples from 5 mice collected at d2, 3 and 4 after $1 \times 10^6$ PFU (blue circles) IV injection of SFV4(A774st). Additional CSF samples analyzed at d3 after $1 \times 10^7$ PFU injection (gray dots). ND: not detected. Limit of detection: 50 PFU/ml. Each data point represents the result from an individual mouse. **d** Representative

immunostaining of choroid plexus 2 days after $1 \times 10^6$ PFU IV SFV4(A774st) injection. SFV protein staining can be seen in the outer surface of VLDLR-positive epithelial cells (indicated with arrows). **e** Representative immunostaining of SFV protein in ventricle wall 2 days after $1 \times 10^6$ PFU IV SFV4(A774st) injection. **f** Representative immunostaining for SFV proteins (green) in WT mouse brain at day 3 after $1 \times 10^6$ PFU IV SFV4(A774st) injection show expression of SFV proteins in ventral regions. Top right corner shows schematic picture of CSF fluid flow in the mouse brain. Created in BioRender. Martikainen, M. (2024) https://BioRender.com/o42e737. All experiments are done with 7–9-week-old female C57BL/6J mice. Source data for (**a**, **c**) is provided as a Source Data file.

*Vldlr* KO mice do not show any SFV staining in brain at d4 (Fig. 4d). In contrast, virus antigen are clearly observed in the brains of wild type B6129SF2/J mice (Fig. 4e). Virus could be expanded from CSF samples from B6129SF2/J and C57BL6/J mice (Fig. S8), which indicates the SFV4(A774st) have access to the CSF in both of these *Vldlr* wild-type mouse models. In line with increased survival and the immuno-fluorescence staining, replicative virus was only detected in one of the *Vldlr* KO mouse brain samples (olfactory bulb, day 4, Fig. 4f), while B6129SF2/J mice have a clear presence of virus in all brain regions at d4 (Fig. 4g). *Vldlr* KO mice also do not show any replicative virus in the CSF samples after IV injection of either $1 \times 10^6$ or $1 \times 10^7$ PFU dose of the virus (Fig. 4f). The blood viremia in *Vldlr* KO is relatively low (Figs. 3e and 4f), which is likely evidence of a generally lower infectivity also in possible target tissues outside the CNS. Despite this, the drastically reduced neuropathology of the VLDLR-independent SFV4(A774st) strain was unexpected and indicates that VLDLR plays a non-redundant role in the

CNS entry of IV injected SFV. As brain endothelial cells do not express VLDLR (Fig. S3), deficiencies in these cells cannot explain the lack of CNS entry in *Vldlr* KO mice. This suggests that brain endothelial cells are therefore neither a highly effective nor a crucial entry route for SFV into the CNS.

To study if *Vldlr* KO mice are completely resistant to SFV4(A774st) we administered the virus directly into the brain (intracranial, IC) or intranasally (IN). Both administration routes led to rapid neuropathology in *Vldlr* KO mice (Fig. 4h), indicating that *Vldlr* KO mice are not generally protected against SFV, and that both virus strains have the potency to replicate if they enter the brains of *Vldlr* KO mice. Intranasal injection (Fig. 4h) led to infection of the olfactory epithelial layers (Fig. 4i), followed by virus reaching not only the olfactory bulb but also deeper brain tissue including the medulla oblongata (Fig. 4j). This points out that unlike CNS entry of SFV from the circulation, the entry of SFV from nasal cavity

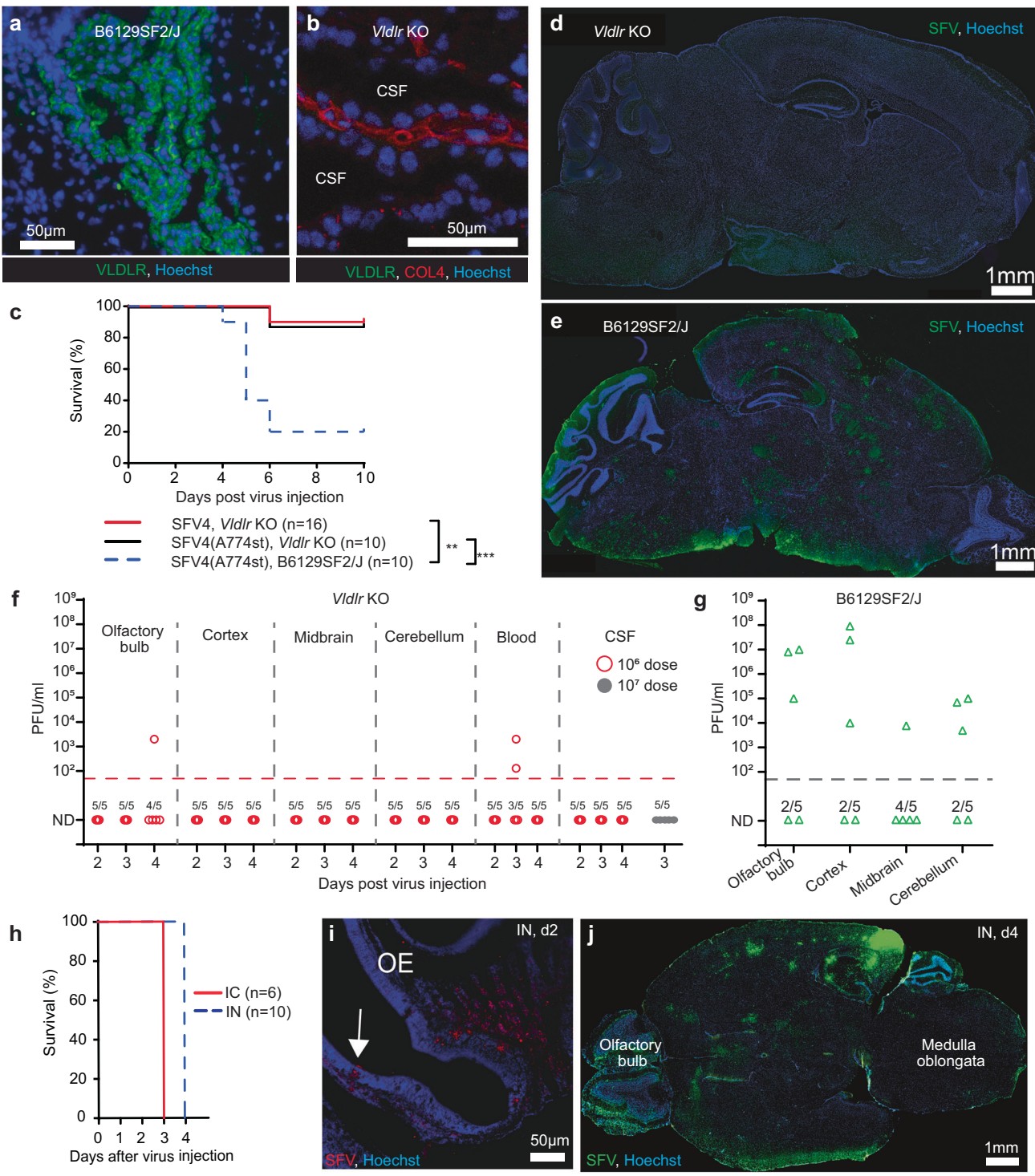

and olfactory epithelium (likely olfactory neurons) is VLDLR-independent in mice.

Taken together, we conclude that VLDLR specifically facilitates SFV neuroinvasion from the circulation, and that lack of VLDLR alone is sufficient to protect mice against intravenous SFV, even if the strain of the used SFV shows non-VLDLR-restricted tropism in vitro. A schematic illustration of the route of SFV neuroinvasion based on our results is presented in Fig. 5.

## Discussion

The identity of the entry receptor for SFV has been a longstanding quest in virology. Our data extends the results by Clark et al.[10], validating VLDLR as an SFV entry receptor. More importantly, we demonstrate that VLDLR expression in the choroid plexus epithelial cells correlates with SFV-mediated neuroinvasion and neuropathology. Thus, SFV binding to ApoER2 (also identified by Clark et al.[10]) cannot compensate for the lack of VLDLR. Our results further

**Fig. 4 | VLDLR is crucial for SFV neuroinvasion from the circulation.** Immuno-fluorescence staining for VLDLR in the choroid plexus of *Vldlr* KO (**a**) and B6129SF2/J (Vldlr wild-type control mouse) mice (**b**). No VLDLR can be detected in *Vldlr* KO sample. CSF: cerebrospinal fluid. **c** Kaplan–Meier analysis of *Vldlr* KO mice infected intravenously (IV) with $1 \times 10^6$ PFU SFV4 or SFV4(A774st) and B6129SF2/J infected intravenously (IV) with $1 \times 10^6$ PFU SFV4(A774st). Statistical analysis done with log-rank test. Data from two independent experiments. Representative immunostaining for SFV proteins in *Vldlr* KO (**d**) and B6129SF2/J (**e**) mouse brain at day 4 after $1 \times 10^6$ PFU IV SFV4(A774st) injection. **f, g** Analysis of virus in different brain regions, blood and CSF with plaque titration (PFU/ml). Samples from 5 *Vldlr* KO mice collected at d2, 3, and 4 after $1 \times 10^6$ PFU (red circles) IV injection of SFV4(A774st). Additional CSF samples analyzed at d3 after $1 \times 10^7$ PFU injection (gray dots). ND: not detected.

**g** plaque titration of brain samples collected from B6129SF2/J mice at d4 after $1 \times 10^7$ PFU SFV4(A774st) injection. Limit of detection: 50 PFU/ml. Each data point represents the result from an individual mouse. **h** Kaplan–Meier analysis of *Vldlr* KO mice infected intracranially (IC) or intranasally (IN) SFV4(A774st). All mice succumb to neurological symptoms. Data from two independent experiments. **i** Representative immunostaining of *Vldlr* KO mouse olfactory epithelium (OE) 2 days after IN SFV4(A774st) injection. SFV staining can be seen at the surface layer of epithelium (indicated with arrow) but also in deeper tissues. **j** Representative immunostaining of *Vldlr* KO mouse brain (transverse plane) 4 days after IN SFV4(A774st) injection. SFV staining can be seen in olfactory bulb but also deeper in the brain tissues. Experiments done with 7–10 week-old female mice. Source data for (**c**, **f**, **g**, **h**) and *P* values for (**c**) are provided as a Source Data file.

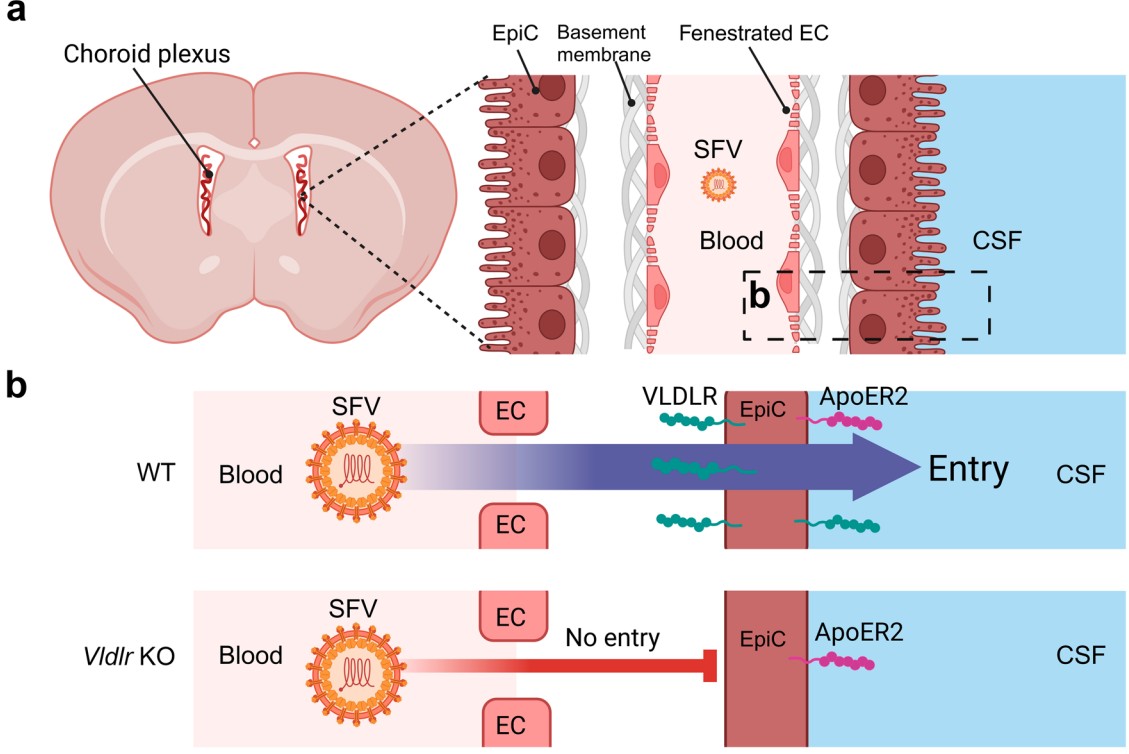

**Fig. 5 | Refined model for SFV neuroinvasion through the blood-CSF barrier.**
**a** Schematic presentation of the choroid plexus and the blood-CSF barrier.
**b** Schematic presentation of VLDLR and ApoER2 expression in endothelial and epithelial cell layers and the effect knocking out *Vldlr* has on generalized SFV entry. Entry through the epithelial layer is solely mediated by VLDLR on the basolateral side, making *Vldlr* KO mice completely resistant to SFV neuroinvasion. Due to the strictly apical expression pattern, ApoER2 does not affect SFV passage through the epithelial cell layer from the basolateral side. Abbreviations: EC: endothelial cell, EpiC: epithelial cell, SFV: Semliki Forest virus, CSF: cerebrospinal fluid, WT: wild-type, KO: knock-out. Created in BioRender. Martikainen, M. (2024) https://BioRender.com/z30c241.

point out that A774 is more capable of VLDLR-independent entry than SFV4. These SFV strains are not interchangeable and differences in their structural spike glycoproteins E1 and E2 results in notable difference in their VLDLR-dependency. SFV4 and A774 structural glycoproteins have previously been shown to bind heparan sulphate with a different affinity[9]. In this context, it is expected that they would also show different affinity to VLDLR or other possible entry receptors.

Unlike VLDLR, which is widely expressed, ApoER2 is preferentially expressed in the CNS[20]. ApoER2-binding could therefore explain SFV neurotropism. The lack of ApoER2 as a hit in our screen can be explained by the strong VLDLR-dependency of the SFV4 strain that was used in the screen. Due to the nature of the screening (i.e., knocking out one gene per cell), the use of more promiscuous SFV constructs such as A774 would not be expected to lead to the discovery of any significant receptor hits.

We additionally identified OR10T2 (Olfactory Receptor Family 10 Subfamily T Member 2) as an alternative receptor candidate. While further validation of this alternative receptor hit is needed, it could explain the effectiveness of intranasal SFV delivery also in *Vldlr* KO mice. While most of the other enriched gRNAs in our screen target genes that encode proteins linked to endosomal escape or translation of viruses, some more interesting hits also can be identified. SFPQ (splicing factor, proline-glutamine rich), a dsRNA-associated factor, is a proviral factor also for Sindbis virus infection[21]. The transcriptional coactivator HCFC1 (host cell factor-1) is critical for the function of herpes simplex virus immediate early enhanceosome complex[22], but has not been shown to be important for alphavirus replication. UBL5 (ubiquitin-like protein 5), SNIP1 (Smad Nuclear Interacting Protein 1), TWISTNB (RNA Polymerase I Subunit F) and MFAP1 (Microfibril Associated Protein 1) have not been reported in affecting viral infection before. Of the identified anti-viral genes, IFNAR1 and STAT2 can be easily explained as being major

components of antiviral IFN-I signaling pathway. Similarly, RAB7A has been reported to be needed for production of IFN-I in HSV-1 infected mouse embryonic fibroblasts (MEFs) by facilitating endosomal trafficking of TLR3[23]. VPS28 is a component of the ESCRT-I (endosomal sorting complex required for transport I) and has been shown to reduce Foot-and-mouth disease virus replication in PK-15 cells[24]. Neither RAB7A nor VPS28 have been reported as anti-viral against alphaviruses before. AQR is a component of spliceosome and POLE2 (DNA poly ε-B subunit) is involved in DNA repair and replication. Neither of these have been directly linked to anti-viral functions so far. Further validation of these hits in our CRISPR library screen is beyond the scope of the currently presented work.

The prevailing model for SFV infection prior to our report has been that SFV enters the CNS through the BBB[8]. However, the lack of VLDLR expression in brain endothelial cells[16] is likely to form a major barrier for VLDLR-dependent SFV clones such as SFV4. Other encephalitic alphaviruses have been reported to utilize transcytosis to pass through the endothelial cells in a receptor-independent manner[25]. Hower, it is currently unknown if SFV could utilize transcytosis for CNS entry. Our results from *Vldlr* KO mice indicate that VLDLR-independent entry mechanisms cannot compensate for the lack of VLDLR in vivo. It should be noted that, while the initial dose of virus into the circulation is the same in both WT and Vldlr KO mice, the lower peripheral replication (due to lack of VLDLR also in peripheral tissues) could contribute to the reduced SFV neuro-invasiveness in *Vldlr* KO mice. Nevertheless, our results indicate that VLDLR is required for the neurovirulent capacity of SFV. The importance of VLDLR is also supported by results from Palakurty et al. 2024, which was published during the revision of our work, that also show reduced SFV pathogenicity in *Vldlr* KO mice[26].

Another feature of the BBB that may further limit the entry of SFV is the structural barrier formed by astrocyte end feet. Given that SFV has poor infectivity in astrocytes[4], it is reasonable to argue that BBB would not be an ideal CNS entry route for SFV. Instead, CSF could serve a more effective route into the brain parenchyma. We can detect SFV in the CSF samples of wild-type mice but the results are relatively inconsistent between different samples. This is likely due to the technical challenge of sample collection and very small sample volumes collected. Further access inside the brain could be facilitated by ependymal cells and/or tanycytes as well as interneurons in perivascular space. Neural progenitor cells and neuroblasts in the subventricular zone could also serve as a way for SFV to penetrate deeper into brain tissue. The possible role of these cells during SFV neuroinvasion requires further investigation.

The blood-CSF barrier as a viral entry route into the CNS has been relatively poorly studied, but has been reported for Zika virus[27] and SARS-CoV-2[28]. Based on our results, we introduce a refined model of SFV neuroinvasion through the blood-CSF barrier. In our model, the passage through the endothelial cell layer can occur by infection or by passive diffusion through fenestrated endothelial cells (Fig. 5). Passage through the epithelial cell layer of the choroid plexus is strictly dependent on VLDLR, which (unlike ApoER2) is also present on the basolateral side of these cells (Fig. 5). In contrast to the BBB, the fenestrated endothelial layer and high VLDLR expression in the epithelial layer of the choroid plexus makes it an ideal gateway for SFV. While the results in Vldlr KO mice are clear and indicate a specific role of choroid plexus epithelial cells in SFV resistance, the conclusion could be strengthened by utilizing a mouse model with a choroid plexus specific *Vldlr* KO. Further studies are also warranted to determine if SFV can utilize circumventricular organs, which have permeable capillaries and direct access to neural tissues, as access points.

Taken together, our results show that SFV neuroinvasion occurs through the B-CSF barrier and that VLDLR is crucial for SFV infectivity in choroid plexus epithelial cells. This indicates a defined and specific role for VLDLR in regulating CNS entry of SFV in mice.

## Methods

All the key resources used in the experiments are listed in Table S1.

### Cell line and viruses

HOS (ATCC, CRL-1543) and K562 (ATCC, #CCL-243) cells were cultured in Gibco RPMI-1640 (Thermo Fisher Scientific, #21875-034) supplemented with 10% Gibco heat-inactivated FBS (Thermo Fisher Scientific, #10500-064), 10 U/mL penicillin-streptomycin (Thermo Fisher Scientific, #15140-122), and 1 mM sodium pyruvate (Thermo Fisher Scientific, #11360-039). To create hVLDLR-expressing K562 cell line, K562 were transduced by lentivirus expressing GFP and VLDLR (pLV[Exp]-EGFP-EF1A>hVLDLR[NM_003383.5], VectorBuilder). Control cells were transduced with lentivirus expressing GFP and Firefly luciferase[29]. After transduction the cells were sorted based on GFP expression using BD FACSMelody cell sorter.

Fully replicative viruses SFV4 (pCMV-SFV4)[30], wtA7(74)[30] and SFV4-d1EGFP[31], A774-mCherry and SFV4-dsRed were obtained from Andres Merits (University of Tartu, Estonia). The chimeric prA774-V4nstr[7] (here denoted SFV4(A774st)) was a kind gift from Ari Hinkkanen (University of Eastern Finland).

Viruses were produced in BHK-21 by transfecting virus plasmid with Lipofectamine 3000 (Thermo Fischer Scientific, # L3000001) followed by collection of supernatant (p0 stock) 2 days later. 500 μl of p0 stock was added on added to a confluent T175 flask of BHK-21 cells to produce the p1 stock, which was harvested 24 h later and concentrated by ultracentrifugation through 20% sucrose cushion (2 h, 140,000 × g, 4 °C). The resulting virus pellet was resuspended in Opti-MEM (Thermo Fisher Scientific, #31985062) overnight at +4 °C, aliquoted and titrated by plaque titration on BHK-21 cells. Shortly, 200 μl from serial dilution of virus (prepared in BHK-21 medium, Thermo Fisher Scientific, # 21710-025) was added on BHK-21 cells on 6 well-plate, let to incubate for 1 h at incubator and covered with 0.6% Carboxymethylcellulose (CMC) -containing BHK-21 medium containing. The cell layer was stained two days later with crystal violet to visualize the plaques. Titer was counted as plaque forming units (PFU)/ml.

### CRISPR/Cas9 library screen

Generation of HOS-Cas9/BFP and transduction with lentiviral human genome-wide Brunello CRISPR-library[12] as service by SciLifeLab CRISPR Functional Genomics unit (Karolinska Institute, Stockholm, Sweden). 40,000,000 cells/sample were used in the screen to maintain full coverage of the library. For the screening cells were given either Ruxolitinib alone (10 μM, Selleck Chemicals, #S5243) or Ruxolitinib and SFV4-d1-EGFP (MOI = 50). Two days later the surviving cells were harvested sgRNA expression was analyzed by MAGeCK analysis[32] by SciLifeLab.

### VLDLR blocking experiments

HOS cells were seeded on 96 well-plate (10,000 cells/well). On the next day medium was replaced with 50 μl VLDLR blocking mAbs 1H5, 1H10, 5F3 (GeneTex, #GTX79551, #GTX79552 and #GTX79550) or IgG1 isotype control antibody (Thermo Fisher Scientific, #14-4714-82) diluted in complete RPMI-1640 medium at 100 μg/ml. 2 h later, 50 μl of virus (MOI = 0.1) diluted in complete RPMI-1640 medium was added and cells. 48 h later, cell viability was measured with Roche Cell Proliferation Kit I MTT assay (Merck Millipore, 11465007001) according to manufacturer's instructions.

Alternatively, HOS cells treated with 1H5, 1H10, 5F3 antibodies (as above) were infected with red fluorophore expressing SFV viruses A774-mCherry or SFV4-dsRed (kind gift form prof. Andres Merits, Tartu, Estonia) using MOI 0.1 and the fluorescence intensity was measured with CLARIOstar plate reader and analyzed with MARS software (BMG Labtech).

## SFV replication kinetics in K562 cell line

200 000 K5621-GFP/Fluc or K562-GFP/VLDLR were seeded on 12 well-plate and infected with SFV4, A774 or A774-V4nstr virus (MOI = 0.01) at 1 ml complete RPMI-164 medium. 200 μl of medium sample was collected at indicated time points and virus amount was quantified by plaque titration in BHK-21 cells.

## Infection of mice

Experimental and control animals were co-housed. The mice were housed in a barrier facility at an average temperature of 23 °C and humidity of 45–65%. The dark/light cycle was fixed to 12 h. Mice were randomly allocated to experimental groups. Sex of the mice was not considered in the study. Female mice were chosen for the experiments due to more stable behavior.

For analysis of neurovirulence, adult (>6-week-old) female C57BL/6J (Charles River Laboratories), *Vldlr*-/- (B6;129S7-Vldlr[tm1Her]/J, The Jackson Laboratory) and B6129SF2/J (The Jackson Laboratory) were used. For intravenous (IV) injection, $1 \times 10^6$ plaque-forming units (PFUs, as titrated in BHK-21 cells[5]) of virus was injected into the tail vein in total volume of 100 μl of phosphate-buffered saline (PBS). Intranasal injections of virus ($1 \times 10^6$ PFUs in 10 μl PBS) were done with a pipette into the left nostril under isoflurane anesthesia. Intracranial injections of virus (1000 PFU in 2 μl PBS) were done 1 mm anterior and 1.5 mm right from bregma at 3-mm depth using a Hamilton syringe and stereotactic equipment (AgnTho's). Mice were monitored daily and sacrificed either upon onset of neurological symptoms (ataxia, paralysis, hunched posture or over 20% loss of body weight) or 10 days after virus injection.

## Analysis of SFV replication in different brain regions, blood, and CSF

Adult female WT (C57BL/6J, Charles River Laboratories) or *Vldlr*-/- (B6;129S7-Vldlrtm1Her/J, The Jackson Laboratory) were infected intravenously with SFV4(A774st) ($1 \times 10^6$ PFU in PBS) followed by collection of samples on days 2, 3 and 4 after virus injection. Mice were put under terminal anaesthesia and CSF was collected with small glass capillary via cisterna magna followed by blood collection via cardiac puncture. Immediately after this the mice were perfused with 10 ml PBS through left ventricle and brains were collected. Brains were dissected to smaller pieces corresponding to specific regions olfactory bulb, cerebellum, cortex (including hippocampus, thalamus, and hypothalamus) and midbrain (including pons and medulla regions). Samples were snap-frozen and stored in −20 °C.

Blood and CSF were directly diluted into BHK-21 medium (Thermo Fisher Scientific, #21710-025) and used for plaque titration. Alternatively, CSF samples from mice were pooled and added on BHK-21 cells grown on 6 well plate for amplification of virus. Frozen brain pieces were suspended into 200 μl of OPTI-MEM (Thermo Fisher Scientific, #31985-062) and homogenized with a disposable tissue grinder (Fisher Scientific, # 13236679) inside 1.5 ml tube. The resulting homogenate was centrifuged max speed 5 min at RT, and the supernatant was used for plaque titration.

## Immunostainings

Brains from sacrificed mice were snap-frozen in 2-methylbutane on dry ice, embedded to OCT embedding matrix (VWR Chemicals, # 361603E) and cut into 7 μm sections with a cryostat and mounted on Superfrost Plus microscope slides (Thermo Fisher Scientific, #J1800AMNZ).

Sections were fixed either with methanol or acetone in −20 °C, washed with PBS and then blocked in PBS containing 3% BSA (1 h RT). Primary antibody incubation was done overnight at 4 °C (with antibodies listed in Table S1) with antibodies diluted 1:200 in blocking buffer. Sections were washed with PBS and incubated with fluorophore-conjugated secondary antibodies diluted 1:500 in PBS (listed in Table S1) for 2 h at room temperature. Sections were the

washed in PBS, counterstained with Hoechst 33342 (Sigma-Aldrich, #14533) and mounted using Fluoromount-G (Thermo Fisher Scientific, #00-4958-02). Fluorescent images were captured with a Zeiss Axioimager microscope (Zeiss) or Leica SP8 confocal microscope (Leica Microsystems). At least two sections from three mice deriving from biologically independent experiments were analyzed.

## In vivo BBB permeability assay

Alexa Fluor-555 Cadaverine (10 μg/g; Thermo Fisher Scientific, A30677) was injected into the tail vein of C57BL/6J mice and let circulate for 2 h. After that, mice were anesthetized and perfused with PBS1x followed by 4% PFA. Brains and kidneys were excised and stored for further analysis. Successful injection of cadaverine was verified by examining kidneys of injected animals under a stereomicroscope (Leica, M205FA) equipped with DFC7000T lamp (Leica Microsystems). Whole brain images were taken using a stereomicroscope and then the brains were vibratome sectioned and processed for immunofluorescent staining.

Vibratome sections (80 μm) from PFA-fixed cryoprotected brain tissues were air-dried and permeabilized in PBS containing 0.1% Triton-X100. Sections were blocked in PBS containing 3% BSA and incubated with anti-CD31 and anti-SFV primary antibodies (listed in Table S1) diluted 1:200 in 1.5% BSA/0.05%Triton-X100 in PBS.

Sections were washed in PBS and stained with Alexa Fluor-conjugated secondary antibodies (listed in key resources table). Nuclei were stained with Hoechst 33342, and the sections were mounted using Fluoromount-G. Tile-scans of brain sagittal sections were obtained using a fluorescent microscope Leica DMi8 (10× objective, Leica Microsystems) and high magnification pictures using a confocal microscope Leica SP8 (63x oil immersion objective).

## Isolation and infection of mouse brain endothelial cells (BECs)

Brains of 10 adult female C57BL6/J mice (Charles River Laboratories) were harvested. Meninges were removed by rolling the brains on sterile Whatman paper, after which brains were minced with scalpel blades, pooled together, and suspended in 50 ml of Buffer A (150 mM NaCl, 5 mM KCl, 2 mM $CaCl_2$, 2.6 mM $MgCl_2$, 15 mM Hepes, 1% BSA in water).

Suspension was centrifuged (5 min, 1200 rpm, RT) and the pellet was suspended in equal volume (compared to the volume of the pellet) of 0.75% Collagenase (Type 2, LS004176, Worthington) in Buffer A. Suspension was incubated 1 h at 37 °C after which the digestion was stopped by adding buffer A up to 50 ml.

After centrifugation (10 min at 1200 rpm at 4 °C) the pellet was suspended in 30 ml of 25% BSA in PBS followed by another round of centrifugation (20 min at 2600 rpm at 4 °C). The supernatant, including a layer of myelin on top, was removed and the pellet suspended in 4 ml of Buffer A. 40 μl of 1% collagenase/dispase (Merck Millipore, #10269638001, in buffer A) was added and suspension was incubated 15 min at 37 °C. After this, 4 μl of DNase (Sigma-Aldrich, # D4513, 1 mg/ml Buffer A) was added and the suspension was incubated additional 2 min at 37 °C, gently shaking to resolve any clumps.

Digestion was stopped by adding Buffer A up to 50 ml and centrifuged for 10 min at 1200 rpm at 4 °C. The resulting cell pellet was suspended in EC medium [DMEM (Thermo Fisher Scientific, #41965-039) complemented with 10% FBS (Thermo Fisher Scientific, #10500-064), 50 μg/ml ECGS (Sigma-Aldrich, #E2759), 100 μg/ml Heparin (Sigma-Aldrich, #H3149-KU50) and Pen/Strep (Thermo Fisher Scientific, #15140-122)] and plated on collagen-coated (Sigma-Aldrich, #C3867) 6well plate (split evenly to all wells) and let grow in 37 °C incubator (5% $CO_2$). On the following day, cells were washed with PBS and new EC medium supplemented with 4 μg/ml puromycin was added.

After 2 days of puromycin (Thermo Fisher Scientific, #A1113803) selection (4 μg/ml) cells were infected by adding SFV4, A774 or

A774-V4nstr virus at MOI = 0.01 (diluted to EC medium without puromycin). Medium samples were collected 24 h after infection for virus titration on BHK-21 cells.

### Flow cytometry analysis of immune cells

Brains of infected 8-week-old old C57BL/6J mice were dissected and processed into cell suspension using Mouse Tumor Dissociation (Miltenyi Biotech #130-096-730) and gentleMACS Dissociator (Miltenyi Biotech). Cells were stained with LIVE/DEAD Fixable Aqua Dead Cell Stain Kit (Thermo Fischer Scientific, #L34957) followed by blocking unspecific Fc receptor binding with anti-mouse CD16/CD32 and subsequent staining for surface markers of interest (listed in Table S1) diluted 1:100 in Brilliant Stain Buffer Plus (BD Biosciences, #566385). Stained samples were acquired using CytoFLEX LX (Beckman Coulter) and data were analyzed using FlowJo software version 10.5.3 (FlowJo LLC).

### Statistical analysis

All statistical analysis was performed using GraphPad Prism software.

### Statistics and reproducibility

All experiments have been repeated successfully. Statistical analysis is done comparing biologically independent replicates. In vivo survival data is pooled from two repeated experiments. Mouse tissue titration data is derived from 5 mice/timepoint. For representative micrographs at least two sections from three mice deriving from biologically independent experiments were analysed with similar results.

### Reporting summary

Further information on research design is available in the Nature Portfolio Reporting Summary linked to this article.

## Data availability

Source data are provided with this paper in Source Data file. The sequencing data from the CRISPR/Cas9 knockout screen generated in this study have been deposited in the GEO database under accession code GSE283607. The scRNA-seq data used in Fig. S3a are available in the Database of gene expression in adult mouse brain and lung vascular and perivascular cells[15,16]. [http://betsholtzlab.org/VascularSingleCells/database.html]. The scRNA-seq data used in Fig. 2d is available in Single Cell Portal[33] under study name "A single-cell transcriptomic atlas of the aging mouse brain"[14] [https://singlecell.broadinstitute.org/single_cell/study/SCP263/aging-mouse-brain]. The scRNA-seq data used in Fig. 2g is available in "Single Cell Portal"[33] under study name "The single-nucleus atlas of the developing, adult, and aged mouse brain choroid plexus"[34] [https://singlecell.broadinstitute.org/single_cell/study/SCP1366/choroid-plexus-nucleus-atlas] Source data are provided with this paper.

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

## Acknowledgements

We want to acknowledge Professor Gerald McInerney (Karolinska Institutet, Sweden), Andres Merits (University of Tartu, Estonia) and Ari Hinkkanen (University of Eastern Finland) for providing SFV-constructs and Tina Sarén (Uppsala University) for helping with FACS assay. The BioVis platform of Uppsala University was used to conduct experiments using light microscopy, supported by Jeremy Adler and the BioVis staff. Part of this work was carried out at the SciLifeLab CRISPR Functional Genomics unit at Karolinska Institutet. We acknowledge support by the National Genomics Infrastructure (NGI); the National Academic Infrastructure for Supercomputing in Sweden (NAISS), partially funded by the Swedish Research Council through grant agreement 2022-06725; and the Uppsala Multidisciplinary Center for Advanced Computational Science (UPPMAX). This work was supported by the Knut and Alice Wallenberg Foundation [KAW 2019.0088, A.D and M.E]; the Swedish Cancer Society [190184Pj M.E. and 222241Pj M.E.]; the Swedish Research Council [2019-01326 M.E. and 2023-02232 M.E.]; the Swedish Childhood Cancer Foundation [PR2020-0167 M.E. and PR2023-0103 M.E.] and the Swedish Brain Foundation [FO2024-0302 M.E.]. M.M. is a recipient of a Marie Curie fellowship from the EU (AVITAG, 707093).

## Author contributions

M.M., R.L., I.P., M.R., D.Y., A.D., and M.E. designed the experiments and/or helped with reagents and material. M.M., R.L, I.P, S.B, C.C, A.S., and M.R. conducted the experiments. M.M., A.D., D.Y., and M.E. wrote the paper.

## Funding

## Competing interests

The authors declare no competing interests.

## Ethical approval

The Swedish Work Environment Authority has approved the work with genetic modification of SFV (ID no. 202100-2932 v66a14 [laboratory] and v67a10 [mice]). All experiments regarding modified SFV were conducted under biosafety level 2. The local Animal Ethics Committee in Stockholm (13414/2020 and 04399/2023) approved the animal studies.
