## [Transparent Peer Review file · Nature Communications]

VLDLR mediates Semliki Forest virus neuroinvasion through the blood-cerebrospinal fluid barrier

Corresponding Author: Dr Miika Martikainen

Version 0:

Reviewer comments:

Reviewer #1

(Remarks to the Author)
Review of Martikainen et al

In this study, the authors performed a genome-wide CRISPR/Cas9 loss-of-infection screen using human osteosarcoma cells treated with the JAK inhibitor ruxolitinib and the alphavirus SFV. One of the top 'hits' in their screen was VLDLR, which recently was established by two other groups as a receptor for SFV (Cao et al., 2023; Clark et al., 2022). They then utilized anti-VLDLR blocking mAbs and KO cells to show differential cell viability or fluorescence reporter activity after infection with strains SFV4, A774 (an attenuated SFV4 strain), or A774-V4nstr (chimera encoding SFV4 nonstructural proteins). Since these strains seem to differentially utilize VLDLR in vitro, the authors then tested their pathogenesis in vivo using VLDLR KO and ApoER2 KO mice. In contrast to WT mice, VLDLR KO mice survived lethal SFV intravenous challenge, with little to no infection in the brain. Interestingly, VLDLR KO mice uniformly succumbed to intracranial and intranasal infection, suggesting a role for VLDLR before or at the level of neuroinvasion. ApoER2 KO mice did not show similar protection, with only 40% survival of SFV-infected mice. The authors performed immunostaining on brain tissues to examine possible VLDLR-specific tropism of SFV, including Purkinje cells, choroid plexus epithelial cells, pericytes, and endothelial cells of the blood-brain barrier. The authors conclude that VLDLR mediates alphavirus neuroinvasion through the blood-cerebrospinal fluid barrier by enabling infection of choroid plexus epithelial cells.

Though several of their findings and hypotheses are interesting, parts of the study lack sufficient evidence for the several mechanisms they propose, or refute, for SFV neuropathogenesis (e.g., refuting the BBB as a possible route of SFV neuroinvasion, proposing blood-CSF route of entry into the brain, with VLDLR-dependent basolateral infection, and concurrent ApoER2-mediated infection through the endothelial layer). Moreover, many of the key mechanistic experiments are relegated to the Supplementary Figures. The story would benefit from tempering of conclusions and shifting the more speculative statements from the Results to the Discussion. Additionally, key controls and experimental details are absent in many main and Supplemental Figures, as described further below.

Major Issues.

1. Rigor. (a) The study could be improved with the addition quantitative immunofluorescence data for the colocalization studies. This should be added to Figures 3, S2F, S3C, and S4. (b) There are issues with the statistical analyses. (i) The authors used unpaired t tests in Figures S1C-E, which should be changed to ANOVA with multiple comparisons corrections. (ii) The survival curves (Figure 2 and S2) need a log-rank analysis. Figure S2I and S5B is missing a wild-type control – are these data the same? (iii) The data in Figures 1E-F, 2H-L, and S3A are missing statistical analysis. Also, in all legends, the data should reflect independent biological experiments – this should be denoted clearly in the legends. (iv) The data in Figure S1D are not compelling without seeing the primary flow cytometry plots/histograms. Also, can the authors show the percentage of cells infected rather than just the MFI?

2. There is a disparity in the localization of dsRNA and SFV antigen staining. It is not clear why the authors do not just use SFV antigen staining since their antibody appears to work well. For example, data in panels 3D and E do not match – the SFV antigen looks more apical in staining.

3. CSF viral load data. Since it is inconsistently positive, did the authors perform any controls to ensure there was no blood contamination in these samples (perfusion was done after CSF sampling)?

4. Figure layout. There are three main Figures (and one model Figure) – however, several of the key data is found in Supplemental data. The authors should integrate key Supplemental data in Figure S2 and S3 into main Figures.

5. BBB permeability. The authors argue that the virus does not cross through the BBB in part, because of data in Figure S4 and lack of expression of VLDLR on endothelial cells. (a) The staining data in Figure S4C-F is not compelling and should be repeated with (i) SFV antigen and (ii) much more extensive quantitation [showing a few isolated cells is not adequate]. (b) Also, there still could be a VLDLR-BBB entry route that is mediated through effects on peripheral infection and cytokine-induced changes (e.g., TNF, IL-1, or IL-6) in BBB permeability that allow virus to gain access to the CNS without directly infecting endothelial cells. It is not clear how the authors have ruled out this possibility, since viral load in blood at days 2-4 is so much lower in VLDLR KO mice. Thus, it remains possible that VLDLR is crucial for viremia (which they show), which could allow BBB crossing either by transcytosis or through weakened cell-to-cell junctions associated with inflammation as was seen with VEEV and WEEV, two other encephalitic alphaviruses (Salimi et al., 2020) – this paper should be cited and discussed. (c) In lines 164-165, the authors state, “The BBB is therefore neither a highly effective nor a crucial entry route for SFV into the CNS.” They would need to perform BBB permeability experiments to state this so categorically. In line 222, the authors state there is no evidence that SFV can pass through endothelial cells by transcytosis. Can they provide a reference for this statement? If not, it should be tempered.

6. Limitations. The authors should add a limitations section. Ideally, the studies could be performed with a conditional VLDLR KO mouse to address the importance of expression on choroid plexus epithelial cells (e.g., Lymphotropic papovavirus control region (LPVcr)-Cre or FOXJ1-Cre (Crouthamel et al., 2012; Jang and Lehtinen, 2022)).

Other issues

1. The Introduction is too short. It would be helpful for the authors to provide additional background and framework for the findings in the manuscript. This includes a description of alphavirus envelope glycoprotein structure and receptor interactions, introducing the concept of virus mechanisms of neuroinvasion via the blood-brain barrier and blood-CSF barrier, and describing more about what is known, or not known, about the cellular route of SFV neuroinvasion and tropism (line 35-36). Beginning line 37, the authors should briefly introduce the CRISPR screen that was performed, why it was of interest to test different SFV strains.

2. Are the VLDLR KO and ApoER2 KO mice fully congenic (backcrossed) to C57BL/6? On the Jax website, they appear to be of mixed background. This should be indicated. If not, how mixed are the mice, and could this affect virological or clinical outcomes? The suggested controls from Jax are B6129SF2/J and littermates (het x het crosses), respectively. Also, in the Results and legends, the authors should indicate the age and sex of mice used for each study. While the Methods state >6 weeks, it is unclear how and if the animals were age-matched.

3. Line 51-52. How did the screen focus on entry factors since it was a loss of killing-based screen. For that matter, throughout the paper, unless experiments specifically showing virus entry, mention of this step in the viral lifecycle should be moved to the Discussion.

4. Line 54. Did any of the other ‘hits’ validate? The authors mention OR10T2 in the Discussion.

5. The authors tested only two SFV strains and showed variable dependence on VLDLR. Do they have access to others? Are most VLDLR-dependent? If not, could this affect their conclusions?

6. Figure 1D and S1C. Why do the authors look at cell viability instead of directly measuring SFV infection by either plaque assay, focus-forming assay, or antigen staining and flow cytometry.

7. Lines 82-83. This statement should be restricted to only the cell lines that the authors tested.

8. Figure 2B and E. Why does the SFV antigen staining look so different in different WT mice at 5 dpi even though scale bars are the same? Please include VLDLR KO staining control at 5 dpi. Additional staining controls with either isotype mAb/control probe on infected and naïve mice would be helpful as well. For Figure 2C, please include a magnified inset for the VLDLR KO since some antigen staining still seems apparent, but it is hard to gauge at lower magnification.

9. Line 96, 101, 104, 122, 195. The authors mention neuropathology in many places. However, none of the fluorescence images show evidence of pathology apart from CD45+ cell infiltrates. Analogously, mortality does not necessarily equate to neurovirulence (line 96) without a more direct analysis.

10. Figure 2J-M. Given that many of the samples are negative for infectious virus, is it possible to re-run them for viral RNA by qRT-PCR, which usually is much more sensitive an assay for detecting viral dissemination. Indeed, the authors make a statement (line 122) that the virus “did not enter the CNS” because brain samples [by plaque assay] were negative. What about viral RNA?

11. Figure S2G: This figure only shows VLDLR KO mice and is missing WT infected controls.

12. Line 119-120: '(including tissues outside the CNS)' - where is this data?
13. Line 123. Avoid using "interestingly." Let the reader decide.
14. Figure 3C. It would be helpful to add a magnified insert, so it parallels the A-B panels.
15. Figure S3A. Change the y-axis to log₁₀ scale to match the data in Figure 2.
16. Figure S4A. Do brain endothelial cells express VLDLR in infected mice? Is it possible that inflammation in the periphery (e.g., cytokines) could upregulate this gene?
17. Figure S6B-D: This data is used as a large part of the mechanism of entry discussion. While suggestive, a single immunostaining experiment showing apical ApoER2 polarization in epithelial cells does not seem completely compelling. Are there any orthogonal experiments that can be performed to confirm and extend this finding?
18. Lines 176-177 and Figure 3D. It would be helpful if the authors could add a similar panel in SFV-infected VLDLR KO mice.
19. Line 193. It should read, "VLDLR as an SFV entry receptor."
20. Lines 214-217 seems highly speculative. Do the authors have any supporting data that OR10T2 is an SFV receptor?

LITERATURE CITED.

- Cao, D., B. Ma, Z. Cao, X. Zhang, and Y. Xiang. 2023. Structure of Semliki Forest virus in complex with its receptor VLDLR. *Cell* 186:2208-2218.e2215.
- Clark, L.E., S.A. Clark, C. Lin, J. Liu, A. Coscia, K.G. Nabel, P. Yang, D.V. Neel, H. Lee, V. Brusica, I. Stryapunina, K.S. Plante, A.A. Ahmed, F. Catteruccia, T.L. Young-Pearse, I.M. Chiu, P.M. Llopis, S.C. Weaver, and J. Abraham. 2022. VLDLR and ApoER2 are receptors for multiple alphaviruses. *Nature* 602:475-480.
- Crouthamel, M.H., E.J. Kelly, and R.J. Ho. 2012. Development and characterization of transgenic mouse models for conditional gene knockout in the blood-brain and blood-CSF barriers. *Transgenic Res* 21:113-130.
- Jang, A., and M.K. Lehtinen. 2022. Experimental approaches for manipulating choroid plexus epithelial cells. *Fluids Barriers CNS* 19:36.
- Salimi, H., M.D. Cain, X. Jiang, R.A. Roth, W.L. Beatty, C. Sun, W.B. Klimstra, J. Hou, and R.S. Klein. 2020. Encephalitic Alphaviruses Exploit Caveola-Mediated Transcytosis at the Blood-Brain Barrier for Central Nervous System Entry. *mBio* 11:

Reviewer #2

(Remarks to the Author)

Martikainen report on studies examining the role of the Very Low-Density Lipoprotein Receptor (VLDLR) in the neuroinvasion of Semliki Forest virus (SFV), a neurotropic alphavirus. This study indicates that choroid plexus (CP) cells at the blood-cerebrospinal fluid barrier (BCSFB) may become infected by SFV and that VLDLR is strictly required for the SFV infection of CP epithelial cells. However, the authors fall short of demonstrating that the BCSFB is a direct entry route for the neuroinvasion of SFV. Thus, the manuscript is too preliminary for publication.

Figure 1. The investigators state they utilized the janus kinase inhibitor ruxolitinib to enhance killing of virally infected cells. Thus, the surviving cells would not be infected with SFV. How is this useful for detecting entry receptors for this virus? Panel 1C indicates that all genes in A774 strain differ from SFV4; is this the case? The data in this figure confirms previously published results identifying VLDLR as an entry receptor for alphaviruses. As the authors indicate that SFV may use other entry receptors, these should have pursued to advance the field.

Figure 2. The result demonstrating that neuroinvasion after intranasal SFV infection is not blocked in *Vldlr*^{-/-} mice is interesting. Do olfactory sensory neurons become infected? What is the entry receptor for SFV in these neurons? Addressing these questions would be an important advance for the field.

Figure S3, S4, S5. While the entry of SFV into CSF is intriguing, this may be due to spread from serum via alterations in tight junctions at the BBB. The detection of virus within the hippocampus supports this, as this area of the brain is highly susceptible to BBB disruption. The authors need to provide ultrastructural analyses demonstrating direct interaction of SFV within the choroid plexus to conclude that this is an entry route. Also, more time points need to be evaluated to demonstrate that SFV does not invade the CNS via infection of astrocyte endfeet, which can be detected via AQ4 IHC. The authors should also consider that there are regional differences in expression of proteins by endothelial cells and astrocytes. The examination of other receptor hits and generation of double Kos could address these questions.

Figure 3. CP epithelial cell markers include TTR, FOLR1, and PRLR, which should be used to co-localize SFV within these cells. These cells form the blood-cerebrospinal fluid barrier (BCSFB), separating the CSF from the blood, which prevents entry of pathogens and molecules into the brain. The authors need to examine whether this barrier is damaged during SFV

infection, and that virus infects cells abutting the ventricular system. The detection of virus in the CP alone does not indicate that this is an entry route into the brain parenchyma.

Reviewer #3

(Remarks to the Author)

Very Low-Density Lipoprotein Receptor (VLDLR) and Apolipoprotein E receptor 2 (ApoER2) have been identified as entry receptors for Semliki Forest virus (SFV), Sindbis virus (SINV) and Eastern equine encephalitis virus (EEEV). These studies convincingly demonstrate that SFV requires VLDLR for entry into the CNS following systemic infection, as direct introduction through intranasal inoculation resulted in equal neurovirulence in both WT and VLDLR KO mice. In all, this is an interesting and well written manuscript with compelling experimental data and the area of investigation is important to our understanding of the impact of viral infection on degenerative disorders of the CNS. In addition, there are a number of questions that should be addressed by the authors.

1. Figure 1. E,F - Were there no statistical differences between SFV4, A774, and A774-V4nstr in cell culture medium?
2. Figure S2 - It appears that replication of SFV4 was localized largely to cortical and possibly subcortical regions (S2B), whereas A774-V4nstr proteins were detected much more widely throughout the CNS in S2H,J (although less so in the cerebellum). It would be helpful to see a similar whole brain and sagittal distribution of SFV4 for S2B to better support the authors' statement of "presence of viral proteins throughout the brain parenchyma (Fig S2B)" (p.3, line 93). The image in Figure 2E is not very clear in this regard.
3. Figure 2F. Putative infiltrating immune cells should be identified with an appropriate marker.
4. Figure 2H-M. Statistical significance should be indicated between groups.
5. Figure S4C-F. It would be helpful to show a merged image of dsRNA with COL4.
6. Figure S6. Figure legend text is cut off at the end of p. 25.
7. Discussion, Figure S6A, and Figure 4. Expression of Vldlr in astrocytes and pericytes appears to be much lower than in epithelial cells of the choroid plexus. Was expression of SFV detected in astrocytes or in pericytes? The schematic in Figure 4 notes pericytes but these cells are not discussed in the text. Likewise, astrocytes are discussed in the text but not noted in Figure 4. Please clarify the presence of the virus in these cell types in accordance with the proposed entry pathway in Figure 4.

Version 1:

Reviewer comments:

Reviewer #1

(Remarks to the Author)

Review of revised Martikainen et al

The authors performed a genome-wide CRISPR/Cas9 screen using human osteosarcoma cells and identified VLDLR as a receptor for SFV, which confirms work by two other groups as (Cao et al., 2023; Clark et al., 2022). They utilized anti-VLDLR blocking mAbs and KO cells to show differential cell viability or fluorescence reporter activity after infection with strains SFV4, A774 (an attenuated SFV4 strain), or SFV4(A774st) (chimera encoding SFV4 nonstructural proteins). VLDLR KO mice survived lethal SFV4(A774st) intravenous challenge, with little to no infection in the brain, whereas VLDLR KO mice uniformly succumbed to intracranial and intranasal infection, suggesting a role for VLDLR in neuroinvasion. The authors performed immunostaining on brain tissues to examine possible VLDLR-specific tropism of SFV, including choroid plexus epithelial cells, pericytes, and endothelial cells of the blood-brain-barrier (BBB). Based on the staining, infection in the CSF, and temporal delays in BBB permeability, the authors conclude that VLDLR mediates alphavirus neuroinvasion through the blood-cerebrospinal fluid barrier by enabling infection of choroid plexus epithelial cells.

The revised paper is clearly improved. It is more streamlined (with data on ApoER2 KO mice removed) to focus on the question of mechanism of SFV neuroinvasion and the role of VLDLR. Nonetheless, there remains some imprecision in the text, an unclear usage of SFV4(A774st) chimera (which uses both VLDLR-dependent and independent entry pathways) for in vivo studies, missing negative staining controls, and some lack of rigor in quantitative and statistical analysis. These issues need to be remedied before publication.

Major Issues.

1. In vivo studies. (a) All in vivo studies in VLDLR KO mice in Figure 4 used the SFV4(A774st) chimeric virus, which based on data in Figure 1, uses both VLDLR-dependent and independent entry pathways. Why were these experiments not performed with SFV4, which more dominantly uses VLDLR? The choice of virus is not justified (and is even confusing based on the statement in lines 201-204).

2. Negative staining controls. Throughout the paper, the authors do not include negative staining controls for VLDLR (using VLDLR KO mice) and SFV (using uninfected mice). These should be added to the Main or Supplementary Figures.

3. Rigor. (a) In some the legends (e.g., Figures 1D, S2C), it appears that the statistical analysis is performed from data of two independent experiments. Most journals (including Nature brand journals) require three independent experiments for such analysis; moreover, the analysis should be performed on mean values from independent biological repeats and not technical repeats. This should be clearly indicated. (b) Many Figure panels (e.g., Figure 3A, 3C-E, 4C, 4G, 4H) do not indicate the number of independent experiments. This should be added. (c) For all representative staining images, the authors need to indicate how many sections were examined, from how many animals, and from how many independent experiments.

Minor comments.

1. p.2, line 26. The sentence has “in mice” used twice. Delete one of them.

2. pp.3-4, lines 90-92. What is the sequence identity in the E1-E2 proteins for the SFV strains. How can the authors claim VLDLR is a “general” receptor based on the study of two strains?

3. p.4, lines 112-114, and Figure 1E. The authors state that VLDLR expression results in “clearly increased virus titers in the culture medium.” How can they make this statement without showing an eclipse phase time point (e.g., 2-3 hours post-infection). What is the increase being compared to?

4. p.5, line 129, and Figure 2G. Is the *Vldlr* and *Lrp8* mRNA expression in EpiCs really different? Can they say it is lower? Based on what analysis?

5. p.5, line 138. Can the authors add a summarizing statement about the similarities and differences in CPeIC staining for VLDLR and ApoER2 (e.g., location/amount)?

6. Figures S4. (a) The authors should add analysis of immune cells from a negative control (uninfected mice); (b) the authors need to indicate these cells in the brain are harvested from perfused mice.

7. Figure 3C. In the legend, please indicate that the numbers at the bottom of the graphs (e.g., 5/5) indicate those at the limit of detection.

8. Figure 3E. Did the higher inoculating dose (10^7) lead to earlier infection in the brain regions at D+2 also (or D+3 samples in other regions) — or just the CSF?

9. p.6, line 165. Please include some description of cadaverine.

10. p.6, line 176 and Figure 4A-B. Where is the data with purebred C57BL/6 mice? It seems as if all the data is with B6129SF2/J mice.

11. p.6, line 180 and 187. Please avoid using the word “drastic” – it is not needed.

12. Figure 4F. Where is the control data [in the CSF] for the viral doses in the B6129SF2/J mice?

13. Figure 4G. Scatter data of individual mice should be shown. The pooling of data (as it appears from the legend – from 5 B6129SF2/J mice) is not appropriate.

14. p.7, lines 191-192. The authors state, “This suggests that brain endothelial cells are therefore neither a highly effective nor a crucial entry route for SFV into the CNS.” Would it be more accurate to state, “brain endothelial cells are not infected in a VLDLR-dependent manner?”

15. p.7, lines 209-210. The authors state, “we demonstrate that VLDLR expression in the choroid plexus epithelial cells is crucial for SFV-mediated neuroinvasion and neuropathology.” Since no conditional KO studies are performed, perhaps a more conservative “correlates with” rather than “is crucial” would be more accurate.

16. p.7, lines 211-212. The authors state, “Our results further point out an important difference in VLDLR utilization between the SFV strains A774 and SFV4.” This statement is likely based on anti-VLDLR antibody inhibition studies in Figure 1. However, the *in vivo* data with the chimeric virus in Figure 4 show that the A774 structural proteins can use VLDLR. So, what exactly is the important difference – it does not seem related to VLDLR but rather the VLDLR-independent entry pathway.

17. p.8, lines 243-244. The authors remark, “However, the lack of VLDLR expression in brain endothelial cells is likely to form a major barrier for VLDLR-dependent SFV clones such as SFV4.” What about transcytosis? Do these cells need to become infected to participate? Also, couldn't VLDLR infection by SFV in peripheral organs impact BBB permeability and allow entry and then crossing into CSF (through systemic cytokines and effects on endothelial cells). Could this also explain why neuroinvasion is not seen in VLDLR KO mice? These points should be raised in the Discussion.

18. p.9, line 259. The authors state, "The blood-CSF barrier as a viral entry into the CNS..." What does this mean? There is a syntax error that needs correction.

19. p.9, line 283. The statement, "Their implication to SFV infection should be therefore considered only as preliminary indication" undercuts the reliability of the screen. Perhaps just delete this statement?

20. Figure 1E-F. Flow cytometry plots or Western blot images should be shown confirming VLDLR expression in K562 cells.

LITERATURE CITED.

Cao, D., B. Ma, Z. Cao, X. Zhang, and Y. Xiang. 2023. Structure of Semliki Forest virus in complex with its receptor VLDLR. *Cell* 186:2208-2218.e2215.

Clark, L.E., S.A. Clark, C. Lin, J. Liu, A. Coscia, K.G. Nabel, P. Yang, D.V. Neel, H. Lee, V. Brusic, I. Stryapunina, K.S. Plante, A.A. Ahmed, F. Catteruccia, T.L. Young-Pearse, I.M. Chiu, P.M. Llopis, S.C. Weaver, and J. Abraham. 2022. VLDLR and ApoER2 are receptors for multiple alphaviruses. *Nature* 602:475-480.

Reviewer #2

(Remarks to the Author)

The authors have not adequately addressed my concerns. Specifically, the main finding in this manuscript is the use of VDLR as a SFV entry receptor, which is not novel, and the authors have not convincingly demonstrated a new route of neuroinvasion for this virus.

Reviewer #3

(Remarks to the Author)

These studies convincingly demonstrate that SFV requires VLDLR for entry into the CNS following systemic infection, as direct introduction through intranasal inoculation resulted in equal neurovirulence in both WT and VLDLR KO mice. In all, this is an interesting and well written manuscript with compelling experimental data and the area of investigation is important to our understanding of the impact of viral infection on degenerative disorders of the CNS. The work is of significance to the field, as viral infection of the CNS is now increasingly linked to long-term neurodegenerative outcomes, although little is known with regard to etiological mechanisms. This study adds significant new data to the literature in this field. Methodology and rigor are sound, and the authors have satisfactorily addressed reviewer comments.

Version 2:

Reviewer comments:

Reviewer #1

(Remarks to the Author)

Review of Martikainen

The authors have performed several additional experiments and controls and addressed the major concerns of this Reviewers. The paper is now improved and will be of great interest to the field. I have only a few very minor suggestions.

1. In Figure S6 (and line 187), why is there staining for VLDLR in cerebellum of *Vldlr*^{-/-} mice? I presume this is background. If so, it should be stated more clearly.

2. Line 100. Should it be with "a" rather than "the" monoclonal antibody

3. Line 268. "neurovirulent" should be "neurovirulent."

4. Line 277. "ZIKA" should be "Zika".

5. Since their original submission, a paper has been published showing reduced SFV mortality in *Vldlr*^{-/-} mice (Palakurty et al. 2024). This publication should be cited and mentioned in the Discussion as it supports their data on the importance of VLDLR to SFV pathogenies.

REFERENCES

Palakurty, S., S. Raju, A. Sariol, Z. Chong, N. Wagoner, H. Ma, O. Zimmerman, L. J. Adams, C. Carmona, Z. Liu, D. H. Fremont, S. P. J. Whelan, W. B. Klimstra, and M. S. Diamond. 2024. 'The VLDLR entry receptor is required for the pathogenesis of multiple encephalitic alphaviruses', *Cell Rep*, 43: 114809.

Response to reviewers

Reviewer #1 (Remarks to the Author):

Review of Martikainen et al

In this study, the authors performed a genome-wide CRISPR/Cas9 loss-of-infection screen using human osteosarcoma cells treated with the JAK inhibitor ruxolitinib and the alphavirus SFV. One of the top 'hits' in their screen was VLDLR, which recently was established by two other groups as a receptor for SFV (Cao et al., 2023; Clark et al., 2022). They then utilized anti-VLDLR blocking mAbs and KO cells to show differential cell viability or fluorescence reporter activity after infection with strains SFV4, A774 (an attenuated SFV4 strain), or A774-V4nstr (chimera encoding SFV4 nonstructural proteins). Since these strains seem to differentially utilize VLDLR in vitro, the authors then tested their pathogenesis in vivo using VLDLR KO and ApoER2 KO mice. In contrast to WT mice, VLDLR KO mice survived lethal SFV intravenous challenge, with little to no infection in the brain. Interestingly, VLDLR KO mice uniformly succumbed to intracranial and intranasal infection, suggesting a role for VLDLR before or at the level of neuroinvasion. ApoER2 KO mice did not show similar protection, with only 40% survival of SFV-infected mice. The authors performed immunostaining on brain tissues to examine possible VLDLR-specific tropism of SFV, including Purkinje cells, choroid plexus epithelial cells, pericytes, and endothelial cells of the blood-brain-barrier. The authors conclude that VLDLR mediates alphavirus neuroinvasion through the blood-cerebrospinal fluid barrier by enabling infection of choroid plexus epithelial cells.

Though several of their findings and hypotheses are interesting, parts of the study lack sufficient evidence for the several mechanisms they propose, or refute, for SFV neuropathogenesis (e.g., refuting the BBB as a possible route of SFV neuroinvasion, proposing blood-CSF route of entry into the brain, with VLDLR-dependent basolateral infection, and concurrent ApoER2-mediated infection through the endothelial layer). Moreover, many of the key mechanistic experiments are relegated to the Supplementary Figures. The story would benefit from tempering of conclusions and shifting the more speculative statements from the Results to the Discussion. Additionally, key controls and experimental details are absent in many main and Supplemental Figures, as described further below.

Thank you, we acknowledge your comprehensive review and very insightful comments. Based on these and remarks from the other two reviewers, we have made a major revision to the manuscript structure and images. All major changes are marked in red in the revised manuscript.

Please, find our point-to-point response to the issues raised in the section below.

Major Issues.

1. Rigor. (a) The study could be improved with the addition quantitative immunofluorescence data for the colocalization studies. This should be added to Figures 3, S2F, S3C, and S4.

We acknowledge that the lack of quantitative IF data is a weakness in our work.

As suggested by you and other reviewers we have included analysis of BBB breakage with quantitative IF data (revised Figure S5). We think that the quantification of cadaverin leakage gives a robust and functionally relevant measurement of BBB infection in mice.

The figures in our revised manuscript have gone through major revisions (according to your major issue comment 4). We have done this with aim to improve the readability and to better communicate the point of our work. The revision includes also removing some microscope pictures (lacking quantification) that were previously presented. We feel that these changes are in line with comments by you and the other reviewers and improve the manuscript.

The removed figures include Figure 3E, F, G, H (please also see our response to your “Other issues” comment 8). We also decided to remove Figures S 2E-F as they are not in the focus of the work. Figure S3 (now Figure 3I) is meant to be qualitative (as do revised Figures 3F and G) and not meant to show colocalization with any other markers.

Getting a quantified overall view of the VLDLR protein expression in different subtypes of endothelial cells and in different brain locations would be technically very challenging. For now, we have chosen to show some IF stainings (revised Figure S3B) as a support for the very poor endothelial cell Vldlr expression as found in three different scRNAseq datasets (revised Figures 2D,G and S3A).

(b) There are issues with the statistical analyses.

(i) The authors used unpaired t tests in Figures S1C-E, which should be changed to ANOVA with multiple comparisons corrections.

Thank you for noticing this. We agree and have changed statistic in Figure S1C-E to ANOVA with multiple comparison correction.

(ii) The survival curves (Figure 2 and S2) need a log-rank analysis.

Thank you for pointing this out. We have added log-rank analysis to all survival curves.

Figure S2I and S5B is missing a wild-type control – are these data the same?

We apologize for mistakenly including the same data panels twice in the manuscript. We have re-evaluated the usability of Lrp8 KO mice and decided to remove this data from the manuscript. Please find our reasoning for this data redaction in our response to your comment “other issues 2”.

(iii) The data in Figures 1E-F, 2H-L, and S3A are missing statistical analysis. Also, in all legends, the data should reflect independent biological experiments – this should be denoted clearly in the legends.

We apologize for the missing statistics. Statistical analysis has been added to graphs when applicable. All the data points which are shown in the manuscript come from biological replicates. This is now clearly stated in the figure legends.

(iv) The data in Figure S1D are not compelling without seeing the primary flow cytometry plots/histograms. Also, can the authors show the percentage of cells infected rather than just the MFI?

The data in Figure S1D (S2D in the revised manuscript) is produced with a plate reader. Hence quantification of percentage of infected cells is unfortunately not possible. We apologize for the confusion and have added text to make the choice of method clear in the revised manuscript.

2. There is a disparity in the localization of dsRNA and SFV antigen staining. It is not clear why the authors do not just use SFV antigen staining since their antibody appears to work well. For example, data in panels 3D and E do not match – the SFV antigen looks more apical in staining.

We agree with your valuable comment, and after careful considerations have decided to remove dsRNA staining figures from the manuscript all together and base our conclusions only to SFV antigen staining as this is more reliable.

3. CSF viral load data. Since it is inconsistently positive, did the authors perform any controls to ensure there was no blood contamination in these samples (perfusion was done after CSF sampling)?

This is an important question. We are collecting the CSF samples very carefully and trying to avoid any blood contamination. After long training we believe that we are able to do so as the samples appear very clear (Response Figure 1A) which suggests that samples are clean CSF. In samples with blood contamination blood is clearly visible (as in Response Figure 1B) are such samples are not used in analysis.

We have noticed that perfusion prior to CSF sampling results in much smaller sample volumes. As the sample volumes are very small (1-3ul), we have chosen to perfuse the mice after CSF sampling.

Response Figure 1. Photographs of CSF samples collected from punctured cisterna magna. A shows clear CSF fluid and B shows sample with obvious blood contamination.

4. Figure layout. There are three main Figures (and one model Figure) – however, several of the key data is found in Supplemental data. The authors should integrate key Supplemental data in Figure S2 and S3 into main Figures.

We agree with your comment and have integrated parts of Figure S2 with new Figure 4. Figure S3 is integrated with new Figure 3.

5. BBB permeability. The authors argue that the virus does not cross through the BBB in part, because of data in Figure S4 and lack of expression of VLDLR on endothelial cells. (a) The staining data in Figure S4C-F is not compelling and should be repeated with (i) SFV antigen and (ii) much more extensive quantitation [showing a few isolated cells is not adequate].

We agree, and have added staining against SFV antigen and quantified BBB leakage (revised Figure S5). We cannot completely exclude the possibility that SFV could also penetrate the BBB which would require (as you point out) much more extensive quantitation of the entire mouse brain. Our conclusion of BBB not being a major entryway into the brain comes from the clear results showing that VLDLR is very low/not expressed in mouse brain endothelial cells, and that VLDLR KO mice are very resistant to IV SFV injections. This indicates that other VLDLR-expressing entry sites must exist. Supporting the choroid plexus as gateway for SFV we show distinctly high VLDLR expression in this structure and evidence of virus infection in choroid plexus at very early timepoint.

(b) Also, there still could be a VLDLR-BBB entry route that is mediated through effects on peripheral infection and cytokine-induced changes (e.g., TNF, IL-1, or IL-6) in BBB permeability that allow virus to gain access to the CNS without directly infecting endothelial cells. It is not clear how the authors have ruled out this possibility,

Thank you for raising an important point. We have analyzed the BBB permeability after SFV infection by using cadaverine. Results have been added to new Figure S6.

Our results indicate SFV protein in endothelial cells and SFV-induced BBB breakage, which would indicate that BBB could become permissive to SFV at later time points during the infection in mice. Our data however points out that initial CNS entry takes place through the B-CSF barrier and that this happens before BBB breakage. We believe that this experiment further strengthens our hypothesis.

since viral load in blood at days 2-4 is so much lower in VLDLR KO mice. Thus, it remains possible that VLDLR is crucial for viremia (which they show), which could allow BBB crossing either by transcytosis or through weakened cell-to-cell junctions associated with inflammation as was seen with VEEV and WEEV, two other encephalitic alphaviruses (Salimi et al., 2020) – this paper should be cited and discussed.

We thank you for the relevant reference. We have cited the work in the revised manuscript discussion.

(c) In lines 164-165, the authors state, “The BBB is therefore neither a highly effective nor a crucial entry route for SFV into the CNS.” They would need to perform BBB permeability experiments to state this so categorically.

We agree with your comment and have added BBB permeability studies into the revised manuscript Figure S6. Our results show that BBB permeability in C57BL6/J mice increases but only at late time-point during SFV pathogenesis. This together with earlier detection of virus in choroid plexus and the resistance of Vldlr KO mice (which cannot be explained by endothelial cells) leads us to conclude that B-CSF is much more crucial for SFV neuroinvasion from the circulation.

In line 222, the authors state there is no evidence that SFV can pass through endothelial cells by transcytosis. Can they provide a reference for this statement? If not, it should be tempered.

As requested, we have now tempered the statement in the revision to “, it is currently unknown if SFV could utilize transcytosis for CNS entry”.

6. Limitations. The authors should add a limitations section. Ideally, the studies could be performed with a conditional VLDLR KO mouse to address the importance of expression on choroid plexus epithelial cells (e.g., Lymphotropic papovavirus control region (LPVcr)-Cre or FOXJ1-Cre (Crouthamel et al., 2012; Jang and Lehtinen, 2022)).

We apologize that this was missing from the manuscript and thank you for the great suggestion for an optimized experimental set up. A limitations section has now been added into the revised manuscript.

Other issues

1. The Introduction is too short. It would be helpful for the authors to provide additional background and framework for the findings in the manuscript. This includes a description of alphavirus envelope glycoprotein structure and receptor interactions, introducing the concept of virus mechanisms of neuroinvasion via the blood-brain barrier and blood-CSF barrier, and describing more about what is known, or not known, about the cellular route of SFV neuroinvasion and tropism (line 35-36). Beginning line 37, the authors should briefly introduce the CRISPR screen that was performed, why it was of interest to test different SFV strains.

We apologize for the shortness of the introduction. We have revised the introduction to provide more background for the current work. We have also added more detailed explanation of the CRISPR screen and the relevance of the used SFV strains.

2. Are the VLDLR KO and ApoER2 KO mice fully congenic (backcrossed) to C57BL/6? On the Jax website, they appear to be of mixed background. This should be indicated. If not, how mixed are the mice, and could this affect virological or clinical outcomes? The suggested controls from Jax are B6129SF2/J and littermates (het x het crosses), respectively. Also, in the Results and legends, the authors should indicate the age and sex of mice used for each study. While the Methods state >6 weeks, it is unclear how and if the animals were age-matched.

You are correct that the C57BL/6J is not a suggested control for the VLDLR KO mice. We have therefore repeated some of the key experiments with B6129SF2/J mice (data presented in revision Figure 4). The data from these mice completely support our earlier conclusion.

We apologize for the vague indication of mouse age and sex. This has been corrected in the revised manuscript. Although all the mice are not exactly age matched (this a practical work-related shortcoming), they all fall into general category of adult mice, and have uniform response to the virus.

We have done more analysis on the Lrp8 (the gene encoding for ApoER2) KO mice and discovered that they likely cannot be considered fully lacking ApoER2 expression. The mice have been engineered by “A neomycin resistance cassette replaced exons 17 and 18, which encode most of the

membrane spanning segment and part of the cytoplasmic tail” (<https://www.jax.org/strain/003524>). As this could mean that truncated and secreted ApoER2 variant is expressed, we performed a RT-qPCR analysis of Lrp8 in heterozygous and homozygous Lrp8 Kos (**Response figure 2**). Our analysis indicates that both heterozygous and homozygous KO mice show clear expression Lrp8 with primers targeted to exon 10 with expression even higher when compared to wild-type control. Levels of analyzed transcript as measured with primers targeted to exon 16/17 and 17/18 junctions show reduced and absent expression in heterozygous and homozygous KO mice respectively.

It would be reasonable to speculate that the truncated/secreted ApoER2 in these mice could possibly act as SFV-neutralizing protein via binding to virus surface and blocking entry, thereby blurring the results. While speculative, we cannot rule this out and are hesitant to draw any conclusion from the results at this point. As this is the case, we have decided to remove the ApoER2 results from the revised manuscript.

Response figure 2. Evidence of truncated ApoER2 expressed in Lrp8 KO mouse brain. **A.** (illustration captured from <https://www.mdpi.com/1422-0067/19/10/3090>) showing ApoER2 variants in mouse. Exon organizations presented with exon numbers and corresponding protein domains. Alternatively spliced exons are indicated by V. Domain-structures shown below the gene structure. **B.** RT-qPCR analysis of VLDLR and Lrp8 transcript expression in Lrp8 heterozygous (Het) and homozygous (KO) mouse brain samples. Lrp8 transcript measured with primers specific to exon 10 or indicated exon junctions (16-17 and 17-18). Expression normalized to GAPDH and compared to wild-type mouse. ND: not detected.

3. Line 51-52. How did the screen focus on entry factors since it was a loss of killing-based screen. For that matter, throughout the paper, unless experiments specifically showing virus entry, mention of this step in the viral lifecycle should be moved to the Discussion.

While your point raised is fair, it is very difficult to dissect the virus entry and replication since detection of entry is based on detection of replication. We do however consider it fair to assume that e.g. using antibodies that bind to cell surface would specifically affect entry. The same applies if the identified or overexpressed genes encode proteins localize to plasma membrane.

4. Line 54. Did any of the other 'hits' validate? The authors mention OR10T2 in the Discussion.

While validation of the other hits is in our interests. We feel that is beyond the scope of our current manuscript. We hope that this can be accepted.

5. The authors tested only two SFV strains and showed variable dependence on VLDLR. Do they have access to others? Are most VLDLR-dependent? If not, could this affect their conclusions?

SFV4 and A774 are the two prototype strains (as better explained in our revised manuscript introduction) and were therefore selected for our studies. They also represent the vast majority of SFV used in published literature. We are interested in finding out if SFV strains/clones that could invade CNS of VLDLR KO mice exist, or which kind of mutations in the virus spike would relate to increased CNS infiltration in VLDLR KO mice. This work is however in a very preliminary stage.

6. Figure 1D and S1C. Why do the authors look at cell viability instead of directly measuring SFV infection by either plaque assay, focus-forming assay, or antigen staining and flow cytometry.

Thank you for the question. SFV is very cytotoxic in a range of cell cultures in vitro. Cell viability assay therefore acts as a good proxy for virus replication. Even so, we have complemented the cell viability assay by also using fluorescent reporter virus and readout with plate reader (revised Figure S2). We have also analyzed produced virus from infected cells with plaque assay (Figure 1F). We see that these measurements together give reliable picture of the viral replication phenotype in vitro.

7. Lines 82-83. This statement should be restricted to only the cell lines that the authors tested.

We have tempered/restricted the statement in the revised manuscript.

8. Figure 2B and E. Why does the SFV antigen staining look so different in different WT mice at 5 dpi even though scale bars are the same?

We are thankful that you pointed this out. There was a mistake in the Figure 2E. Here the time point should say d4. We apologize the mistake and have corrected the figure. We have revised our microscope pictures (including the scale bars) and provide higher quality images in the revised manuscript.

Please include VLDLR KO staining control at 5 dpi. Additional staining controls with either isotype mAb/control probe on infected and naïve mice would be helpful as well. For Figure 2C, please include a magnified inset for the VLDLR KO since some antigen staining still seems apparent, but it is hard to gauge at lower magnification.

Our SFV antibody is polyclonal anti-serum collected from infected rabbits (gift from Ari Hinkkanen) and used successfully in previous publications (eg. PMID: 33869741 and PMID: 28557974). The staining is reliable as controlled by staining of infected brain section at different time points and naïve mice. Please find secondary antibody control staining of mouse choroid plexus region in the attached Response Figure 3.

The dsRNA antibody we used in the previous version of the manuscript (Figures 3E-H and S4C-F) was produced in mouse. As seen in the figure anti-mouse secondary antibody gives some amount of unspecific staining (can be variable in different samples) increasing chance of false positive staining when using dsRNA antibody. Because of this and the reliability of anti-SFV antibody, we have decided to remove the dsRNA staining images from the revised manuscript.

We apologize the poor quality of Figures 2B and C. These have been replaced with better quality images (revised Figures 3B and 4D respectively). Unfortunately, we do not have a VLDLR KO sample harvested at d5. The revised figure 4D shows image of VLDLR KO brain at day 4 (d4) and should be compared to the image panel 4E which shows B6129SF2/J mouse brain also at d4.

Response Figure 3. Secondary antibody control staining in mouse choroid plexus. Host species indicated with each panel. E is magnification from D and shows unspecific staining in choroid plexus structure.

9. Line 96, 101, 104, 122, 195. The authors mention neuropathology in many places. However, none of the fluorescence images show evidence of pathology apart from CD45+ cell infiltrates. Analogously, mortality does not necessarily equate to neurovirulence (line 96) without a more direct analysis.

Your point that mortality not necessarily equating to neurovirulence is correct. However, in the case of SFV the model for virus induced encephalitis is well established by a body of previously published work. The mice start developing symptoms rapidly after virus injection and when sacrificed have virus and CD45+ cell infiltrate in the brain. We also show BBB damage associated with the SFV infection (revised Figure S6) these findings are all in line with previously established SFV neurovirulence.

10. Figure 2J-M. Given that many of the samples are negative for infectious virus, is it possible to re-run them for viral RNA by qRT-PCR, which usually is much more sensitive an assay for detecting viral dissemination. Indeed, the authors make a statement (line 122) that the virus “did not enter the CNS” because brain samples [by plaque assay] were negative. What about viral RNA?

Thank you for the comment. We have now analyzed the same samples that were used for titration with RT-qPCR using primers specific for viral genome. This type of analysis can be very sensitive, but quantification and normalization are difficult, especially if accurate number of viral genome copy numbers is wanted. Analysis (even if done in accuracy of viral genome copy number) can be difficult to correlate into quantity of infectious viral particles. In this respect, the plaque assay easily gives a usable and comparable estimation of functional virus titers in the tissue.

Comparison of RT-qPCR and titration shows that the results are comparable (response Figure 5A, B).

We also cannot reliably detect (CT values < 30) virus with RT-qPCR from VLDLR KO samples which show no presence of virus with plaque assay (Response Figure 3C and D). The only exception is the d3 olfactory bulb sample, which shows PCR positivity at day 3 (Response Figure 5C) when titration shows negative result (revised manuscript Figure 4F).

Response Figure 4. RT-qPCR analysis of virus in brain samples. A. PCR analysis of virus in B6129SF2/J mouse brain samples at d 3 after SFV4/A774 IV injection. Results correlate with titer obtained by plaque assay (B). C and D, PCR analysis of Vldlr KO mouse brain samples collected at d3 (C) or d4 (D) after SFV4/A774 IV injection. All other samples except olfactory bulb show clear PCR positivity (Ct value < 30).

11. Figure S2G: This figure only shows VLDLR KO mice and is missing WT infected controls.

Correct. The aim of this graph (revised Figure 4H) together with IF staining (revised Figure 4I and J) is to show that virus can enter Vldlr KO mice if administered IN, not to compare survival to WT mice. We have tried to clarify this in the manuscript.

12. Line 119-120: '(including tissues outside the CNS)'- where is this data?

This is a speculative statement that we have tempered in the revised manuscript to "possible target tissues outside the CNS".

13. Line 123. Avoid using "interestingly." Let the reader decide.

Suggestion well taken, the word "interestingly" is not used in revised manuscript.

14. Figure 3C. It would be helpful to add a magnified insert, so it parallels the A-B panels.

Thank you for the suggestion. This panel is moved to Figure 4B in the revised manuscript and is magnified.

15. Figure S3A. Change the y-axis to log₁₀ scale to match the data in Figure 2.

Thank you for pointing out this mistake that has slipped us. The scale is now changed in the revised manuscript.

16. Figure S4A. Do brain endothelial cells express VLDLR in infected mice? Is it possible that inflammation in the periphery (e.g., cytokines) could upregulate this gene?

Thank you for raising an intriguing hypothesis. While this is possible, we have not observed this in our IF stainings.

According to available literature, expression of VLDLR is driven by PPARs (at least in mouse adipocytes, PMID:21924248). PPAR signaling is generally associated with anti-inflammatory response with reported negative correlation between proinflammatory cytokines and PPAR activity in SARS-CoV-2 patients (reviewed in PMID: 36875108). It could be therefore speculated that increased cytokine concentrations would lead to lower VLDLR expression. Whether this would apply to mouse endothelial cells remains an open question.

17. Figure S6B-D: This data is used as a large part of the mechanism of entry discussion. While suggestive, a single immunostaining experiment showing apical ApoER2 polarization in epithelial cells does not seem completely compelling. Are there any orthogonal experiments that can be performed to confirm and extend this finding?

Apologies but it is unclear to us what “orthogonal experiments to confirm or extend the finding” exactly means. Expression of ApoER2 seems to be relatively low (revised Figure 2) and the observed staining localizes to the apical side. This finding is perfectly in line with strictly apical polarization of ApoER2 in choroid plexus in the paper by Stockinger et al. (PMID: 9822699, also cited in the manuscript).

18. Lines 176-177 and Figure 3D. It would be helpful if the authors could add a similar panel in SFV-infected VLDLR KO mice.

We have revised the manuscript and reorganized the pictures. We cannot see SFV staining or replicative virus (with the exception of one olfactory bulb sample) in VLDLR KO mice.

19. Line 193. It should read, “VLDLR as an SFV entry receptor.”

Thank you, this has been revised in the new manuscript.

20. Lines 214-217 seems highly speculative. Do the authors have any supporting data that OR10T2 is an SFV receptor?

Unfortunately, at this point further validation of OR10T2 is beyond the scope of our manuscript. We have indicated this in the Limitations part of the revised manuscript.

LITERATURE CITED.

Cao, D., B. Ma, Z. Cao, X. Zhang, and Y. Xiang. 2023. Structure of Semliki Forest virus in complex with its receptor VLDLR. *Cell* 186:2208-2218.e2215.

Clark, L.E., S.A. Clark, C. Lin, J. Liu, A. Coscia, K.G. Nabel, P. Yang, D.V. Neel, H. Lee, V. Brusic, I. Stryapunina, K.S. Plante, A.A. Ahmed, F. Catteruccia, T.L. Young-Pearse, I.M. Chiu, P.M. Llopis, S.C. Weaver, and J. Abraham. 2022. VLDLR and ApoER2 are receptors for multiple alphaviruses. *Nature* 602:475-480.

Crouthamel, M.H., E.J. Kelly, and R.J. Ho. 2012. Development and characterization of transgenic mouse models for conditional gene knockout in the blood-brain and blood-CSF barriers. *Transgenic Res* 21:113-130.

Jang, A., and M.K. Lehtinen. 2022. Experimental approaches for manipulating choroid plexus epithelial cells. *Fluids Barriers CNS* 19:36.

Salimi, H., M.D. Cain, X. Jiang, R.A. Roth, W.L. Beatty, C. Sun, W.B. Klimstra, J. Hou, and R.S. Klein. 2020. Encephalitic Alphaviruses Exploit Caveola-Mediated Transcytosis at the Blood-Brain Barrier for Central Nervous System Entry. *mBio* 11:

Reviewer #2 (Remarks to the Author):

Martikainen report on studies examining the role of the Very Low-Density Lipoprotein Receptor (VLDLR) in the neuroinvasion of Semliki Forest virus (SFV), a neurotropic alphavirus. This study indicates that choroid plexus (CP) cells at the blood-cerebrospinal fluid barrier (BCSFB) may become infected by SFV and that VLDLR is strictly required for the SFV infection of CP epithelial cells. However, the authors fall short of demonstrating that the BCSFB is a direct entry route for the neuroinvasion of SFV. Thus, the manuscript is too preliminary for publication.

Thank you, we appreciate your insightful commentary on our manuscript. We have prepared a comprehensive revision to the manuscript to improve the readability and added data to strengthen our conclusions. All major changes are marked in red in the revised manuscript.

Please, find our point-to-point response to your comments below.

Figure 1. The investigators state they utilized the janus kinase inhibitor ruxolitinib to enhance killing of virally infected cells. Thus, the surviving cells would not be infected with SFV. How is this useful for detecting entry receptors for this virus?

We agree with you that the experimental design of the CRISPR screen requires better explanation in the manuscript and have therefore added text to the revised manuscript to explain the effect of ruxolitinib and added the result of screen without ruxolitinib to illustrate the effect.

Shortly, the detection of enriched gene hits is dependent on the death of the infected target cells. These target cells can resist the virus either in the entry or replication step. As the entry precedes the replication, the inhibited replication could then rescue some cells even though the virus was able to enter.

Ruxolitinib inhibit type I interferon (IFN-I) signaling through the JAK/STAT pathway. IFN-I induced anti-viral response is a major mechanism with which the target cells can attenuate viral replication. From experience, we know that the IFN-I driven antiviral state can be triggered by lentiviral transduction thereby making lentivirally engineered cell artificially resistant to SFV. Ruxolitinib removes this issue at the same time as it skews the screening more towards discovery of genes which are crucial of SFV entry.

Panel 1C indicates that all genes in A774 strain differ from SFV4; is this the case?

Thank you for the question. A774 and SFV4 derive from different original virus isolates from African mosquitoes. They share notable similarity but also sequence differences in every gene. This is why we have chosen to indicate them with different colors.

The data in this figure confirms previously published results identifying VLDLR as an entry receptor for alphaviruses. As the authors indicate that SFV may use other entry receptors, these should have pursued to advance the field.

We agree that identification of alternative receptors should be pursued to get a complete view of SFV entry. We have preliminary data indicating that some olfactory receptor(s) could also function as SFV receptors but cannot validate this at this point. The advance we offer to the field with this manuscript is the specific and crucial role of VLDLR in vivo and its role in neurotoxicity cause by SFV.

Figure 2. The result demonstrating that neuroinvasion after intranasal SFV infection is not blocked in Vldlr^{-/-} mice is interesting. Do olfactory sensory neurons become infected? What is the entry receptor for SFV in these neurons? Addressing these questions would be an important advance for the field.

We agree that this is an interesting result and acknowledge you sharing this view. We can observe SFV infection both in the outer layer of olfactory epithelium and deeper in the tissue (revised manuscript Figure 4I and J). Unfortunately, we cannot at this point identify the specific type of the infected cells or which receptor SFV uses to enter these cells. The question is interesting but beyond the scope of our current manuscript.

Figure S3, S4, S5. While the entry of SFV into CSF is intriguing, this may be due to spread from serum via alterations in tight junctions at the BBB. The detection of virus within the hippocampus supports this, as this area of the brain is highly susceptible to BBB disruption. The authors need to provide ultrastructural analyses demonstrating direct interaction of SFV within the choroid plexus to conclude that this is an entry route.

We assume that your request of ultrastructural analysis means using electron microscopy to visualize virus particles inside choroid plexus. We agree that this would increase the certainty of our findings. However, it would be very challenging in practice. At this point, we can already show infection in the choroid plexus epithelial cells (revised Figure 3F) which very clearly indicates that the virus has entered the choroid plexus. The direction of virus infection would also be hard to decisively conclude from the electron microscope pictures.

Also, more time points need to be evaluated to demonstrate that SFV does not invade the CNS via infection of astrocyte endfeet, which can be detected via AQ4 IHC. The authors should also consider that there are regional differences in expression of proteins by endothelial cells and astrocytes. The examination of other receptor hits and generation of double KOs could address these questions.

Thank you for the comment. The time scale of SFV-induced neuropathology is relatively short. In this time scale our current work is focused on the early CNS infiltration. Here we cannot detect SFV in astrocytes.

Your point of possible regional differences in SFV receptor expression in the brain is certainly intriguing and of interest to us. However, at this point it is beyond the reach of our work.

Similarly, further *in vivo* validation of other receptor hits remains the topic of our future work. Double KO (if meaning Vldlr and Lrp8) has a severe underdevelopment in the brain (mainly cerebellum) and therefore mice are not viable for experimentation.

Figure 3. CP epithelial cell markers include TTR, FOLR1, and PRLR, which should be used to co-localize SFV within these cells.

Thank you for the comment. The choroid plexus can be easily recognized by the anatomical structure. Similarly, we can differentiate the epithelial cell layer with collagen IV staining which clearly shows the basement membrane between the endothelial and epithelial cell layers. According to the data in our manuscript, VLDLR is distinctly highly expressed in the CP epithelial cells, therefore also serving as marker for these cells.

SFV protein is clearly detected on the apical surface of choroid plexus (revised Figure 3G) which consists of epithelial cells. As you pointed out, we also see FOLR1 staining in CP epithelial cells (response Figure 5).

Response Figure 5. FOLR1 staining in C57BL6/J mouse choroid plexus.

These cells form the blood-cerebrospinal fluid barrier (BCSFB), separating the CSF from the blood, which prevents entry of pathogens and molecules into the brain. The authors need to examine whether this barrier is damaged during SFV infection, and that virus infects cells abutting the ventricular system. The detection of virus in the CP alone does not indicate that this is an entry route into the brain parenchyma.

Thank you, analysis of choroid plexus damage during SFV infection is a good suggestion. Despite our efforts, we have not been able to detect that SFV-virus cause direct damage in the choroid plexus. At the endpoint we do see leakiness in the BBB (Figure S5) and substantial immune cell infiltration (Figure S4). The immune cell infiltration can be seen all around the brain and is likely to play a major contribution to the overall brain damage during infection. The immune-induced and virus infection-caused damage are difficult to differentiate as they could also be expected to appear around the same time after virus appearing inside the brain. On a related note, we do not know how resistant the choroid plexus epithelial cells are to SFV cytotoxicity.

Your comment on cells abutting the ventricular system is very relevant. In the revised manuscript Figures 3G-I, we show infection in the ventricle wall and brain locations that would match with natural flow of CSF.

We agree that detection of virus alone does not indicate that this is an entry route into the CNS. Our conclusion is based on multiple supporting finding such as time of virus appearing in CP, the drastic effect of Vldlr KO specifically into pathogenicity of intravenous SFV, the lack of vldlr mRNA expression in brain endothelial cells and the distinct expression of VLDLR in the CP epithelial cells.

Reviewer #3 (Remarks to the Author):

Very Low-Density Lipoprotein Receptor (VLDLR) and Apolipoprotein E receptor 2 (ApoER2) have been identified as entry receptors for Semliki Forest virus (SFV), Sindbis virus (SINV) and Eastern equine encephalitis virus (EEEV). These studies convincingly demonstrate that SFV requires VLDLR for entry into the CNS following systemic infection, as direct introduction through intranasal inoculation resulted in equal neurovirulence in both WT and VLDLR KO mice. In all, this is an interesting and well written manuscript with compelling experimental data and the area of investigation is important to our understanding of the impact of viral infection on degenerative disorders of the CNS. In addition, there are a number of questions that should be addressed by the authors.

Thank you for your valuable comments. Based on your remarks and remarks from the other two reviewers, we have made a major revision to the manuscript. All major changes are marked in red in the revised manuscript.

Please, find our point-to-point response to the questions below.

1. Figure 1. E,F - Were there no statistical differences between SFV4, A774, and A774-V4nstr in cell culture medium?

Thank you for pointing this out. We have now added statistical analysis to the figure.

2. Figure S2 - It appears that replication of SFV4 was localized largely to cortical and possibly subcortical regions (S2B), whereas A774-V4nstr proteins were detected much more widely throughout the CNS in S2H,J (although less so in the cerebellum). It would be helpful to see a similar whole brain and sagittal distribution of SFV4 for S2B to better support the authors' statement of "presence of viral proteins throughout the brain parenchyma (Fig S2B)" (p.3, line 93). The image in Figure 2E is not very clear in this regard.

Thank you. We have tempered the text and revised the figures.

3. Figure 2F. Putative infiltrating immune cells should be identified with an appropriate marker.

As requested, we have added flow cytometry analysis of different immune cell populations (revised Figure S4)

4. Figure 2H-M. Statistical significance should be indicated between groups.

Statistical analysis of these graphs is difficult since most of the values are "ND". We have also split the groups in different figures since C57BL/6 does not represent the genetically correct control strain. We hope that this sufficiently answers the question.

5. Figure S4C-F. It would be helpful to show a merged image of dsRNA with COL4.

Thank you for the comment. After careful consideration and as response to comments by other reviewer, we have decided to remove dsRNA staining from the revised manuscript. Instead, we have analyzed endothelial cell infection by SFV antigen staining. Figure S5C shows merged image of SFV protein and CD31 (endothelial marker).

6. Figure S6. Figure legend text is cut off at the end of p. 25.

We apologize for this and have revised the manuscript legends.

7. Discussion, Figure S6A, and Figure 4. Expression of Vldlr in astrocytes and pericytes appears to be much lower than in epithelial cells of the choroid plexus. Was expression of SFV detected in astrocytes or in pericytes? The schematic in Figure 4 notes pericytes but these cells are not discussed in the text. Likewise, astrocytes are discussed in the text but not noted in Figure 4. Please clarify the presence of the virus in these cell types in accordance with the proposed entry pathway in Figure 4.

Thank you for your comments. We agree that this part could be presented more clearly and revised the Figure 5 and the discussion text.

Pericytes are part of the B-CSF barrier. It is correct that although we included these cells in the schematic picture, we do not have much evidence on if SFV infect these cells or not. Since pericytes have low expression of Vldlr and Lrp8 (revised Figures 2D and S3A) they likely are resistant to SFV infection. Similarly, we do not have indication of SFV infecting astrocytes. The resistance of astrocytes is also well established in previously published papers.

Although pericytes are present in the choroid plexus, we have now removed them from our illustration in figure 5. Astrocytes are not a component of the B-CSF barrier and therefore not depicted in the figure.

REVIEWER COMMENTS

Reviewer #1 (Remarks to the Author):

Review of revised Martikainen et al

The authors performed a genome-wide CRISPR/Cas9 screen using human osteosarcoma cells and identified VLDLR as a receptor for SFV, which confirms work by two other groups as (Cao et al., 2023; Clark et al., 2022). They utilized anti-VLDLR blocking mAbs and KO cells to show differential cell viability or fluorescence reporter activity after infection with strains SFV4, A774 (an attenuated SFV4 strain), or SFV4(A774st) (chimera encoding SFV4 nonstructural proteins). VLDLR KO mice survived lethal SFV4(A774st) intravenous challenge, with little to no infection in the brain, whereas VLDLR KO mice uniformly succumbed to intracranial and intranasal infection, suggesting a role for VLDLR in neuroinvasion. The authors performed immunostaining on brain tissues to examine possible VLDLR-specific tropism of SFV, including choroid plexus epithelial cells, pericytes, and endothelial cells of the blood-brain-barrier (BBB). Based on the staining, infection in the CSF, and temporal delays in BBB permeability, the authors conclude that VLDLR mediates alphavirus neuroinvasion through the blood-cerebrospinal fluid barrier by enabling infection of choroid plexus epithelial cells.

The revised paper is clearly improved. It is more streamlined (with data on ApoER2 KO mice removed) to focus on the question of mechanism of SFV neuroinvasion and the role of VLDLR. Nonetheless, there remains some imprecision in the text, an unclear usage of SFV4(A774st) chimera (which uses both VLDLR-dependent and independent entry pathways) for in vivo studies, missing negative staining controls, and some lack of rigor in quantitative and statistical analysis. These issues need to be remedied before publication.

Thank you for your insightful comments and suggestions. We trust that our revised manuscript addresses all your concerns. Please find our point to point response below.

Major Issues.

1. In vivo studies. (a) All in vivo studies in VLDLR KO mice in Figure 4 used the SFV4(A774st) chimeric virus, which based on data in Figure 1, uses both VLDLR-dependent and independent entry pathways. Why were these experiments not performed with SFV4, which more dominantly uses VLDLR? The choice of virus is not justified (and is even confusing based on the statement in lines 201-204).

Thank you for the comment. We have added justification for the choice of virus as follows:

“We decided to employ SFV4(A774st) for our further studies due to its ability to utilize both VLDLR-dependent and independent entry therefore allowing relevant conclusion to be made about the importance of disrupting VLDLR-dependent entry in vivo “

2. Negative staining controls. Throughout the paper, the authors do not include negative staining controls for VLDLR (using VLDLR KO mice) and SFV (using uninfected mice). These should be added to the Main or Supplementary Figures.

Thank you for pointing this out. We have added negative staining controls as requested into supplementary Figure S6.

3. Rigor. (a) In some the legends (e.g., Figures 1D, S2C), it appears that the statistical analysis is performed from data of two independent experiments. Most journals (including Nature brand journals) require three independent experiments for such analysis; moreover, the analysis should be performed on mean values from independent biological repeats and not technical repeats. This should be clearly indicated.

Thank you for the comment. We have repeated the experiments with statistics to allow correct analysis (as suggested). This is also now clearly indicated in the figure legends.

(b) Many Figure panels (e.g., Figure 3A, 3C-E, 4C, 4G, 4H) do not indicate the number of independent experiments. This should be added.

We apologize for not clearly indicating this. This information has been added to the Figures.

(c) For all representative staining images, the authors need to indicate how many sections were examined, from how many animals, and from how many independent experiments.

We apologize for overlooking this. We have added this information to the revised manuscript Materials and methods "Immunostainings"-section as follows: "At least 2 sections from 3 mice deriving from biologically independent experiments were analyzed."

Minor comments.

1. p.2, line 26. The sentence has "in mice" used twice. Delete one of them.

We apologize for the mistake. The sentence is corrected in the new revision.

2. pp.3-4, lines 90-92. What is the sequence identity in the E1-E2 proteins for the SFV strains. How can the authors claim VLDLR is a "general" receptor based on the study of two strains?

Thank you for pointing this out. We have now added sequence comparison of other commonly cited SFV strains L10, SFV6 and A7 into revised Fig S2A, and revised the text accordingly:

"Sequence alignment with other common SFV strains SFV6, L10, and A7, shows that SFV4 and A774 also capture the general sequence variation observed across these strains (Fig S2A)."

3. p.4, lines 112-114, and Figure 1E. The authors state that VLDLR expression results in "clearly increased virus titers in the culture medium." How can they make this statement without showing an eclipse phase time point (e.g., 2-3 hours post-infection). What is the increase being compared to?

We apologize for the lack of clarity in the statement. We have added statistical analysis to the Figure 1E. and revised the manuscript text as follows:

"The VLDLR-negative K562 cell line supported infection of both A774 and SFV4(A774st) resulting in >100-fold higher virus titers in the culture medium as compared to SFV4 at the d2 time point post infection (Fig. 1E). This further reinforces the lower VLDLR-dependency of A774. In K562 cells engineered to express VLDLR (K562-VLDLR, Fig. 1F), titers of all viruses increased as compared to wild-type K562 cells (6)."

4. p.5, line 129, and Figure 2G. Is the Vldlr and Lrp8 mRNA expression in EpiCs really different? Can they say it is lower? Based on what analysis?

Thank you for the comment. As accurate comparative analysis is from this data is difficult, we have tempered the statement. “Compared to VLDLR, lower percentage CPEpiCs show Lrp8 expression (Fig 2D and 2G).”

5. p.5, line 138. Can the authors add a summarizing statement about the similarities and differences in CPEiC staining for VLDLR and ApoER2 (e.g., location/amount)?

Thank you for the good suggestion. We have added the following summarizing statement:

“Taken together, these data suggest that from the two previously reported SFV receptors VLDLR and ApoER2, only VLDLR is abundantly present on the basolateral membrane of CPEpiCs.”

6. Figures S4. (a) The authors should add analysis of immune cells from a negative control (uninfected mice);

Thank you for the comment, we have added analysis of control mice to the Fig S5.

(b) the authors need to indicate these cells in the brain are harvested from perfused mice.

We apologize for not indicating this in the manuscript. The cells in flow analysis were harvested from non-perfused mice. This is also indicated in the revised Figure S5 legend.

7. Figure 3C. In the legend, please indicate that the numbers at the bottom of the graphs (e.g., 5/5) indicate those at then limit of detection.

Thank you for the comment. We have added limit of detection to the Figure legends.

8. Figure 3E. Did the higher inoculating dose (10^7) lead to earlier infection ion the brain regions at D+2 also (or D+3 samples in other regions) — or just the CSF?

We unfortunately do not have an answer to this question since we did not collect brains for titration from this experiment. Of note, while we can detect more virus in the CSF after higher dose IV injection we do not see the virus appearing earlier.

9. p.6, line 165. Please include some description of cadaverine.

As requested, we have added the following description: “SFV4(A774st) infected C57BL6/J mice show signs of BBB damage, as indicated by leakage of IV injected cadaverine, as a small-molecule (950 Da) fluorescent probe, into the brain parenchyma at day 4 after virus injection (Fig S7A-E).”

10. p.6, line 176 and Figure 4A-B. Where is the data with purebred C57BL/6 mice? It seems as if all the data is with B6129SF2/J mice.

In Figure 4, we are comparing the VLDLR KO to the B6129SF2/J mice because it is the closest genetic control we have. Data from C57BL/6J is presented in Figure 3.

11. p.6, line 180 and 187. Please avoid using the word “drastic” – it is not needed.

The word drastic has been deleted as suggested.

12. Figure 4F. Where is the control data [in the CSF] for the viral doses in the B6129SF2/J mice?

We apologize for missing this data from the manuscript. We have conducted a virus amplification experiment from CSF samples collected from both B6129SF2/J and C57BL/6J mice. Results of this experiment are added to the revised manuscript Figure S8.

13. Figure 4G. Scatter data of individual mice should be shown. The pooling of data (as it appears from the legend – from 5 B6129SF2/J mice) is not appropriate.

Thank you for pointing this out. We have added scatter data of individual mice to the revised manuscript Figure 4G.

14.p.7, lines 191-192. The authors state, “This suggests that brain endothelial cells are therefore neither a highly effective nor a crucial entry route for SFV into the CNS.” Would it be more accurate to state, “brain endothelial cells are not infected in a VLDLR-dependent manner?”

Thank you for the comment. This statement is based on our results and we feel that changing it as suggested would undermine the conclusions of our work.

15. p.7, lines 209-210. The authors state, “we demonstrate that VLDLR expression in the choroid plexus epithelial cells is crucial for SFV-mediated neuroinvasion and neuropathology.” Since no conditional KO studies are performed, perhaps a more conservative “correlates with” rather than “is crucial” would be more accurate.

Thank you for the suggestion. We have revised the text accordingly.

16. p.7, lines 211-212. The authors state, “Our results further point out an important difference in VLDLR utilization between the SFV strains A774 and SFV4.” This statement is likely based on anti-VLDLR antibody inhibition studies in Figure 1. However, the in vivo data with the chimeric virus in Figure 4 show that the A774 structural proteins can use VLDLR. So, what exactly is the important difference – it does not seem related to VLDLR but rather the VLDLR-independent entry pathway.

Thank you for the suggested clarification to the text. We have revised the sentence as follows; “Our results further point out that A774 is better capable of VLDLR-independent entry than SFV4.”

17. p.8, lines 243-244. The authors remark, “However, the lack of VLDLR expression in brain endothelial cells is likely to form a major barrier for VLDLR-dependent SFV clones such as SFV4.” What about transcytosis? Do these cells need to become infected to participate? Also, couldn't VLDLR infection by SFV in peripheral organs impact BBB permeability and allow entry and then crossing into CSF (through systemic cytokines and effects on endothelial cells). Could this also explain why neuroinvasion is not seen in VLDLR KO mice? These points should be raised in the Discussion.

Thank you for the comment. Transcytosis playing a role in neuroinvasion is a possibility but difficult to rule out. It is also reasonable to speculate that transcytosis would be also dependent on virus binding a receptor on cell surface.

Our results show that SFV gains entry into brain parenchyma of WT mice before BBB leakage occurs. This would indicate that SFV neuroinvasion is not dependent on increased BBB permeability.

Of note, the CSF titers in WT mice seem to peak with blood viremia but not with brain titers. This can be taken as evidence of CSF acting as temporary passageway for SFV when moving into the brain, instead of virus ending up into the CSF from the infected brain tissue.

We have revised the discussion to include the following paragraph:

“The prevailing model for SFV infection prior to our report has been that SFV enters the CNS through the BBB 8. However, the lack of VLDLR expression in brain endothelial cells 16 is likely to form a major barrier for VLDLR-dependent SFV clones such as SFV4. Other encephalitic alphaviruses have been reported to utilize transcytosis to pass through the endothelial cells in a receptor-independent manner²⁵. However, it is currently unknown if SFV could utilize transcytosis for CNS entry. Our results from Vldlr KO mice indicate that VLDLR-independent entry mechanisms cannot compensate for the lack of VLDLR in vivo. It should be noted that, while the initial dose of virus into the circulation is the same in both WT and Vldlr KO mice, the lower peripheral replication (due to lack of VLDLR also in peripheral tissues) could contribute to the reduced SFV neuro-invasiveness in Vldlr KO mice. Nevertheless, our results indicate that VLDLR is required for the neurovirulent capacity of SFV. “

18. p.9, line 259. The authors state, “The blood-CSF barrier as a viral entry into the CNS...” What does this mean? There is a syntax error that needs correction.

We apologize for the error. The sentence is corrected “The blood-CSF barrier as a viral entry route into the CNS...”.

19. p.9, line 283. The statement, “Their implication to SFV infection should be therefore considered only as preliminary indication” undercuts the reliability of the screen. Perhaps just delete this statement?

Thank you for the comment. This sentence has been deleted from the revised manuscript.

20. Figure 1E-F. Flow cytometry plots or Western blot images should be shown confirming VLDLR expression in K562 cells.

Thank you for pointing this out. We have added IF staining which confirms the VLDLR expression into Figure 1F.

LITERATURE CITED.

Cao, D., B. Ma, Z. Cao, X. Zhang, and Y. Xiang. 2023. Structure of Semliki Forest virus in complex with its receptor VLDLR. *Cell* 186:2208-2218.e2215.

Clark, L.E., S.A. Clark, C. Lin, J. Liu, A. Coscia, K.G. Nabel, P. Yang, D.V. Neel, H. Lee, V. Brusic, I. Stryapunina, K.S. Plante, A.A. Ahmed, F. Catteruccia, T.L. Young-Pearse, I.M. Chiu, P.M. Llopis, S.C. Weaver, and J. Abraham. 2022. VLDLR and ApoER2 are receptors for multiple alphaviruses. *Nature* 602:475-480.

Reviewer #1 (Remarks to the Author):

Review of Martikainen

The authors have performed several additional experiments and controls and addressed the major concerns of this Reviewers. The paper is now improved and will be of great interest to the field. I have only a few very minor suggestions.

Thank you for your comments. We greatly appreciate the effort you have taken to review our manuscript in detail.

1. In Figure S6 (and line 187), why is there staining for VLDLR in cerebellum of *Vldlr*^{-/-} mice? I presume this is background. If so, it should be stated more clearly.

You are correct. We have now stated (both in text and figure legend) that this is background staining.

2. Line 100. Should it be with “a” rather than “the” monoclonal antibody

Thank you for pointing this out. We have changed this to “a monoclonal antibody”

3. Line 268. “neurovirulent” should be “neurovirulent.”

Thank you for pointing this out. This has been fixed in the revised manuscript.

4. Line 277. “ZIKA” should be “Zika”.

Thank you for pointing this out. We have corrected this in the revised manuscript.

5. Since their original submission, a paper has been published showing reduced SFV mortality in *Vldlr*^{-/-} mice (Palakurty et al. 2024). This publication should be cited and mentioned in the Discussion as it supports their data on the importance of VLDLR to SFV pathogenies.

Thank you for the good suggestion. We have cited this paper in the discussion as follows:

“The importance of VLDLR is also supported by results from Palakurty et al. 2024, which was published during the revision of our work, that also show reduced SFV pathogenicity in *Vldlr* KO mice (Palakurty et al. 2024).”